# Series of Hessian-Vector Products for Tractable Saddle-Free Newton Optimisation of Neural Networks

**Elre T. Oldewage\*** *etv21@cam.ac.uk*
*Department of Engineering*
*University of Cambridge*

**Ross M. Clarke\*** *rmc78@cam.ac.uk*
*Department of Engineering*
*University of Cambridge*

**José Miguel Hernández-Lobato** *jmh233@cam.ac.uk*
*Department of Engineering*
*University of Cambridge*

*(\*Equal contribution; randomly ordered)*

**Reviewed on OpenReview:** `https://openreview.net/forum?id=qBZeQBEDIW`

## Abstract

Despite their popularity in the field of continuous optimisation, second-order quasi-Newton methods are challenging to apply in machine learning, as the Hessian matrix is intractably large. This computational burden is exacerbated by the need to address non-convexity, for instance by modifying the Hessian's eigenvalues as in Saddle-Free Newton methods. We propose an optimisation algorithm which addresses both of these concerns — to our knowledge, the first efficiently-scalable optimisation algorithm to asymptotically use the exact inverse Hessian with absolute-value eigenvalues. Our method frames the problem as a series which principally square-roots and inverts the squared Hessian, then uses it to precondition a gradient vector, all without explicitly computing or eigendecomposing the Hessian. A truncation of this infinite series provides a new optimisation algorithm which is scalable and comparable to other first- and second-order optimisation methods in both runtime and optimisation performance. We demonstrate this in a variety of settings, including a ResNet-18 trained on CIFAR-10.

## 1 Introduction

At the heart of many machine learning systems is an optimisation problem over some loss surface. In the field of continuous optimisation, second-order Newton methods are often preferred for their rapid convergence and curvature-aware updates. However, their implicit assumption of a (locally) convex space restricts their usability, requiring the use of mechanisms like damping (Martens, 2010; Dauphin et al., 2014; O'Leary-Roseberry et al., 2021) to avoid degenerate behaviour. In machine learning applications, which are invariably non-convex, high dimensionality further plagues this class of optimiser by creating intractably large Hessian (second-derivative) matrices and a proliferation of saddle points in the search space (Pascanu et al., 2014). These difficulties constrain most practical systems to first-order optimisation methods, such as stochastic gradient descent (SGD) and Adam.

Pascanu et al. (2014) and Dauphin et al. (2014) tackled some of these challenges by proposing Saddle-Free Newton (SFN) methods. In essence, they transform the Hessian by taking absolute values of each eigenvalue, which makes non-degenerate saddle points repel second-order optimisers where typically they would be attractive. Because this transformation would otherwise require an intractable eigendecomposition of the Hessian, they work with a low-rank Hessian approximation, on which this process is achievable, albeit at the cost of introducing an additional source of error.

In this paper, we propose a new route towards SFN optimisation which exploits Hessian-vector products to avoid explicitly handling the Hessian. We use a squaring and square-rooting procedure to take the absolute value of the eigenvalues without eigendecomposing the Hessian and deploy an infinite series to tractably approximate the expensive square-root and inverse operations. The resulting algorithm is comparable to existing methods in both runtime and optimisation performance, while tractably scaling to larger problems, even though it does not consistently outperform the widely-known Adam (Kingma & Ba, 2015) and KFAC (Martens & Grosse, 2015). To our knowledge, this is the first approximate second-order approach to (implicitly) take the absolute value of the full Hessian matrix's eigenvalues and be exact in its untruncated form. After summarising previous work in Section 2, we mathematically justify the asymptotic exactness of our algorithm in Section 3 and show its practical use in a range of applications in Section 4. Section 5 concludes the paper.

## 2 Related Work

Although stochastic first-order optimisation methods are the bread and butter of deep learning optimisation, considerable effort has been dedicated to *preconditioned* gradient methods – methods that compute a matrix which scales the gradient before performing an update step. Newton's method and quasi-Newton methods, which multiply the gradient by the Hessian or an approximation thereof, fall into this category. Other examples include AdaGrad (Duchi et al., 2011) which calculates a preconditioner using the outer product of accumulated gradients, and SHAMPOO (Gupta et al., 2018) which is similar to Adagrad but maintains a separate, full preconditioner matrix for each dimension of the gradient tensor.

Martens (2010) proposes using *Hessian-Free* (HF) or truncated Newton (Nocedal & Wright, 2006) optimisation for deep learning. The algorithm uses finite differences to approximate the Hessian in combination with the linear conjugate gradient algorithm (CG) to compute the search direction. Like our method, HF implicitly works with the full Hessian matrix and is exact when CG converges.

Pascanu et al. (2014) and Dauphin et al. (2014) present the proliferation of saddle points in high-dimensional optimisation spaces as an explanation for poor convergence of first-order optimisation methods. Various approaches to escaping these saddle points have been proposed. Jin et al. (2017) observe that saddle points are easy to escape by adding noise to the gradient step when near a saddle point, as indicated by a small gradient. Another idea is to normalise the gradient so that progress is not inhibited near critical points due to diminishing gradients (Levy, 2016; Murray et al., 2019).

Saddle points also present a hurdle to second order optimisation, since they become attractive when applying Newton's method. Nevertheless, some work leverages second order information in sophisticated ways to avoid saddle points. For example, Curtis & Robinson (2019) exploit negative curvature information by alternating between classical gradient descent steps and steps in the most extreme direction of negative curvature. Adolphs (2018) builds on this to propose "extreme curvature exploitation", where the eigenvectors corresponding to the most extreme positive and negative eigenvalues are added to the vanilla gradient update step. Anandkumar & Ge (2016) develop an algorithm which finds stationary points with first, second *and third* derivatives equal to zero, and show that progressing to a fourth-order optimality condition is NP-hard. Truong et al. (2021) project the Newton update step onto subspaces constructed using the positive- and negative-curvature components of the Hessian, allowing them to negate the updates proposed by the latter.

Nesterov & Polyak (2006) show that regularising Newton's method with the cubed update norm secures convergence to second-order stationary points, with further developments improving robustness to approximate Hessians (Tripuraneni et al., 2018); considering approximate (Cartis et al., 2011) or efficient (Carmon & Duchi, 2019) solutions to the cubic-regularised sub-problem; introducing momentum (Wang et al., 2020); or extending the analysis to apply to mini-batched settings (Wang et al., 2019). Limiting the parameter updates to some *trust region* in which the Newton approximation can be relied upon provides an additional analytical perspective (Nocedal & Wright, 2006), with sub-solvers such as that of Curtis et al. (2021) having appealing convergence behaviour. However, the latter note "these methods have often been designed primarily with complexity guarantees in mind and, as a result, represent a departure from the algorithms that have proved to be the most effective in practice".

Pascanu et al. (2014) propose the Nonconvex Newton Method, which constructs a preconditioner by decomposing the Hessian and altering it so that all eigenvalues are replaced with their absolute values and very small eigenvalues are replaced by a constant. Unfortunately, explicit decomposition of the Hessian is expensive and does not scale well to machine learning applications. Dauphin et al. (2014) extend this work by proposing the Saddle-Free Newton (SFN) method, which avoids computing and decomposing the exact Hessian by an approach similar to Krylov subspace descent (Vinyals & Povey, 2012), which finds $k$ vectors spanning the $k$ most dominant eigenvectors of the Hessian. However, this approach relies on the Lanczos algorithm, which has historically been unstable (Cahill et al., 2000; Scott, 1979) and requires careful implementation to avoid these issues (Saad, 2011). O'Leary-Roseberry et al. (2021) instead invert a low-rank approximation to the Hessian for improved stability. However, their method is susceptible to poor conditioning at initalisation and is limited to very small step sizes in settings with high stochasticity. Consequently, it is unclear how well the algorithm extends beyond the transfer learning settings illustrated.

Other saddle-avoiding approaches include detecting non-convexity by comparing optimisation performance to that which would be expected if the function were convex. Carmon et al. (2017) do this by analysing the convergence of Nesterov-accelerated gradient descent, while Royer et al. (2020) combine the Conjugate Gradient method with damping to leverage non-convex update steps to the extent allowed by a trust region and (Liu & Roosta, 2023) follows non-convex directions identified by the Minimum-Residual method to escape indefinite regions.

Instead, our work writes the inverse of the squared and principal square-rooted Hessian as a series, of which we can compute a truncation without explicitly computing or eigendecomposing the Hessian, thereby avoiding certain instabilities faced by Dauphin et al. (2014) and O'Leary-Roseberry et al. (2021). This can be seen as an extension of Agarwal et al. (2017)'s use of a Neumann series to approximate the product of the inverse Hessian and a gradient vector.

There are other examples in machine learning where infinite series are used to motivate approximations to the inverse Hessian (Lorraine et al., 2020; Clarke et al., 2022); we exploit the same construction as Song et al. (2021) to compute the square root of a matrix.

An alternative approach is to precondition the gradient with a curvature matrix that is positive semi-definite by definition, thereby circumventing concerns surrounding saddle points. Notably, the natural gradient method (Amari, 1998) preconditions the gradient with the inverse Fisher information matrix, rather than the inverse Hessian. Whereas the Hessian measures curvature in the model parameters, the Fisher quantifies curvature in terms of the KL-divergence between model and data probability distributions. The natural gradient can be approximated by methods like Factorized Natural Gradient (Grosse & Salakhudinov, 2015) and Kronecker-Factored Approximate Curvature (KFAC) (Martens & Grosse, 2015). In particular, KFAC approximates the Fisher with a block diagonal matrix, which significantly reduces the memory footprint and reduces the cost of inversion. KFAC also leverages several other "tricks", which are relevant for later discussion. We provide a brief overview below and further details in Appendix A.4:

**Moving average of curvature matrix** KFAC maintains an online, exponentially-decaying average of the approximate curvature matrix, which improves its approximation thereof and makes the method more robust to stochasticity in mini-batches.

**Adaptive learning rate and momentum factor** KFAC's update rule incorporates a learning rate and a momentum factor which are both computed adaptively by assuming a locally quadratic model and solving for the local model's optimal learning rate and momentum factor at every iteration.

**Tikhonov damping with Levenberg-Marquardt style adaptation.** KFAC incorporates two damping terms: $\eta$ for weight regularisation, and $\lambda$ which is adapted throughout training using Levenberg-Marquardt style updates (Moré, 1978). The damping constant $\lambda$ can be interpreted as defining a trust region for the update step. When the curvature matrix matches the observed landscape, the trust region is grown by shrinking $\lambda$; otherwise damping is increased so that optimisation becomes more SGD-like.

KFAC is arguably the most popular second-order method enjoying widespread use in practice, so we include it as an important baseline in Section 4. Section 4.1 describes our studies incorporating similar adaptive mechanisms into our method, which we now proceed to derive.

## 3 Derivations

Suppose we wish to minimise some scalar function $f(\mathbf{x})$ over the vector quantities $\mathbf{x}$, which have some optimal value $\mathbf{x}^*$. Denote by $\mathbf{g} = \nabla_{\mathbf{x}} f = \nabla f(\mathbf{x})$ and $\mathbf{H} = \nabla_{\mathbf{x}} (\nabla_{\mathbf{x}} f)^{\mathsf{T}}$ the gradient vector and Hessian matrix of $f$, respectively, with both quantities evaluated at the present solution $\mathbf{x}$. We make no assumptions about the convexity of $f(\mathbf{x})$.

### 3.1 Preliminaries

Under a classical Newton framework, we can approximate a stationary point $\mathbf{x}^*$ by writing a second-order Taylor series for perturbations around some $\mathbf{x}$. Assuming $\mathbf{H}$ is invertible, this recovers

$$\mathbf{x}^* \approx \mathbf{x} - \mathbf{H}^{-1}\mathbf{g}, \tag{1}$$

where the RHS is the Newton update to $\mathbf{x}$. In effect, we have locally approximated $f$ about $\mathbf{x}$ by a quadratic function, then set $\mathbf{x}$ to the stationary point of this quadratic. The invertibility of $\mathbf{H}$ guarantees that this stationary point is unique. However, if $\mathbf{H}$ is not positive definite — for instance, if the function is locally non-convex — that stationary point may be a maximum or saddle point of the approximated space, rather than a minimum.

To address this limitation, we might consider the eigendecomposition of $\mathbf{H}$. Since $\mathbf{H}$ is real and symmetric for non-degenerate loss functions, its eigenvalues are real and its eigenvectors may be chosen to be orthonormal. We can interpret the eigenvectors as the 'principal directions of convexity', and the eigenvalues as the corresponding magnitudes of convexity in each direction (where negative eigenvalues encode concavity). As $\mathbf{H}^{-1}$ has equal eigenvectors to $\mathbf{H}$ and reciprocal eigenvalues, we may interpret the product $\mathbf{H}^{-1}\mathbf{g}$ as a transformation of the gradient vector, with anisotropic scaling governed by the directions and magnitudes of convexity in $\mathbf{H}$. Moreover, this product gives *exactly* the updates necessary to move along each principal direction of convexity to the stationary value in that direction, according to the locally quadratic approximation implied by $\mathbf{H}$. This is illustrated in Figure 1.

As positive eigenvalues are associated with directions of convex curvature, $\mathbf{H}^{-1}$ selects updates in these directions which *decrease* the loss function. Conversely, $\mathbf{H}^{-1}$ selects updates which *increase* the loss function in the directions associated with negative eigenvalues — directly opposing our goal of minimising $f$. Intuitively, we would like to reverse the direction of the latter updates, such that they are decreasing $f$. This is equivalent to changing the sign of the corresponding eigenvalues.

This intuitive idea was presented by Pascanu et al. (2014), and Dauphin et al. (2014) establish a more direct derivation using a trust region framework, which motivates taking the absolute value of every eigenvalue in a *Saddle-Free Newton* (SFN) method (Figure 1). However, its implementation in deep learning is challenged by the intractably large Hessian matrices of non-trivial neural networks. Previous work (Dauphin et al., 2014; O'Leary-Roseberry et al., 2021) tackles this by computing a low-rank approximate Hessian, whose eigendecomposition may be calculated directly, changed as required and approximately inverted. While such an approach secures tractability, the cascade of approximations threatens overall accuracy.

### 3.2 Absolute Values as Square-Rooted Squares

Our proposed method seeks to transform the eigenvalues of $\mathbf{H}$ without computing its full eigendecomposition. This approach is inspired by the observation that, for scalar $x$, $|x| = +\sqrt{x^2}$, where we specifically take the positive square root. In the matrix case, we may define $\mathbf{S}$ as a square root of a square matrix $\mathbf{A}$ iff $\mathbf{A} = \mathbf{SS}$. For a square, positive semi-definite $\mathbf{A}$, there is a unique positive semi-definite square root $\mathbf{B}$, which we term the *principal* square root of $\mathbf{A}$; we will write $\mathbf{B} = \sqrt[+]{\mathbf{A}}$.

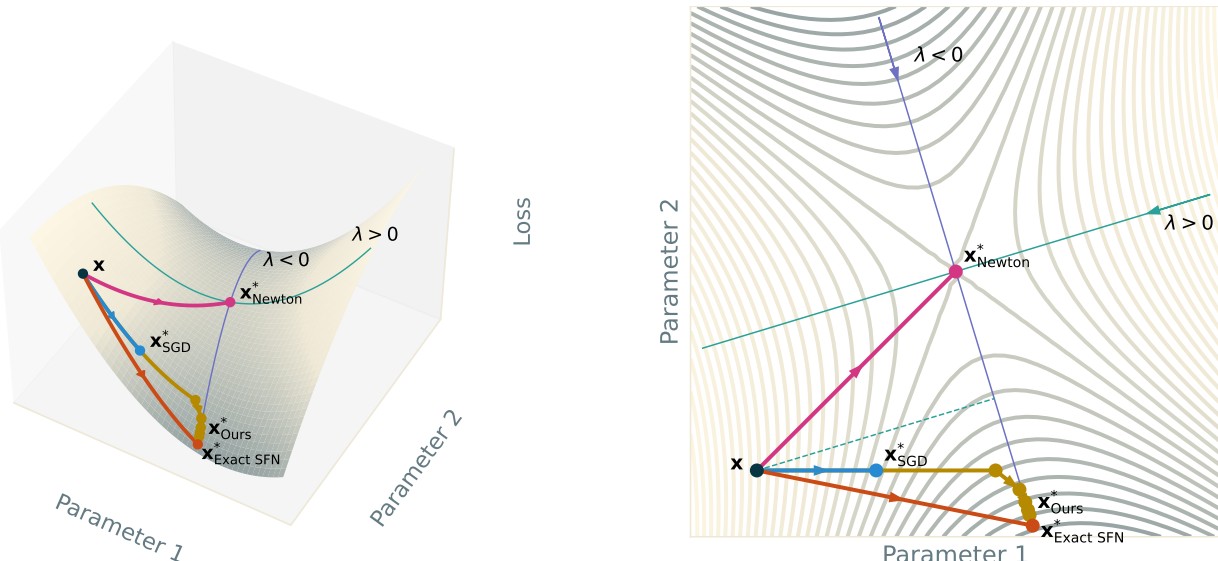

Figure 1: Motivation for Saddle-Free Newton methods. This locally quadratic surface has a saddle point (•) and its Hessian gives two principal directions of curvature (——, ——). From any initial point (•), SGD will give an update neglecting curvature (→•) and Newton's method converges immediately to the saddle point (→•). Exact Saddle-Free Newton (→•) takes absolute values of the Hessian eigenvalues, negating the components of the Newton update in concave directions (——) and thus changing the saddle point from an attractor to a repeller. Our series-based method (→•) is an approximate Saddle-Free Newton algorithm which converges to the exact Saddle-Free Newton result.

If $\mathbf{A}$ is real and symmetric, we may eigendecompose it as $\mathbf{Q\Lambda Q}^\mathsf{T}$, where $\mathbf{Q}$ is the orthonormal matrix whose columns are the eigenvectors of $\mathbf{A}$ and $\mathbf{\Lambda}$ the diagonal matrix whose elements are the corresponding eigenvalues of $\mathbf{A}$. Then, we have $\mathbf{B} = \mathbf{Q\Lambda}^{\frac{1}{2}}\mathbf{Q}^\mathsf{T}$. Since raising the diagonal matrix $\mathbf{\Lambda}$ to the $k$th power is equivalent to raising each diagonal element to the $k$th power, $\mathbf{B}$ has the same eigenvectors as $\mathbf{A}$, but the eigenvalues of $\mathbf{B}$ are the square roots of those of $\mathbf{A}$. By taking the principal square root, we guarantee that all the eigenvalues of $\mathbf{B}$ are non-negative, hence we have taken the positive square root of each eigenvalue in turn.

This reveals a route to transforming our Hessian $\mathbf{H}$ by taking the absolute value of its eigenvalues. Consider the eigendecomposition $\mathbf{H} = \mathbf{Q\Lambda Q}^\mathsf{T}$, noting that $\mathbf{H}^2 = \mathbf{Q\Lambda}^2\mathbf{Q}^\mathsf{T}$ is positive semi-definite by construction, as its eigenvalues are squares of real numbers. But then $\sqrt[+]{\mathbf{H}^2}$ is the unique positive semi-definite square root of $\mathbf{H}^2$, and each eigenvalue of $\sqrt[+]{\mathbf{H}^2}$ is the positive square root of the square of the corresponding eigenvalue of $\mathbf{H}$ — equivalently, its absolute value. Thus, we may take the absolute value of each eigenvalue of $\mathbf{H}$ by computing the square and then taking the principal square root of $\mathbf{H}$, i.e. by computing $\sqrt[+]{\mathbf{H}^2}$.

### 3.3 Inverse Square Root Series

To use this transformed $\mathbf{H}$ as a second-order preconditioner, we must also invert it, so the matrix of interest is $\left(\sqrt[+]{\mathbf{H}^2}\right)^{-1}$. We now develop a series approximation to this quantity. For scalars $z$, we may exploit the generalised binomial theorem to write

$$(1-z)^{-\frac{1}{2}} = \sum_{k=0}^{\infty} \frac{1}{2^{2k}}\binom{2k}{k}z^k \tag{2}$$

Applying the root test for convergence, a sufficient condition for the convergence of this series is $\limsup_{n\to\infty}|z^n|^{\frac{1}{n}} < 1$. We generalise this series to the matrix case by replacing the absolute value $|\cdot|$ with any compatible sub-multiplicative matrix norm $\|\cdot\|$ and writing $\mathbf{I} - \mathbf{Z}$ in place of $1 - z$. Ideally, we would set

$\mathbf{Z} = \mathbf{I} - \mathbf{H}^2$ and recover a power series directly, but to ensure convergence we will require a scaling factor $V$ such that $\mathbf{Z} = \mathbf{I} - \frac{1}{V}\mathbf{H}^2$. With this addition, we have

$$(\mathbf{H}^2)^{-\frac{1}{2}} = \frac{1}{\sqrt{V}} \sum_{k=0}^{\infty} \frac{1}{2^{2k}} \binom{2k}{k} \left( \mathbf{I} - \frac{1}{V}\mathbf{H}^2 \right)^k. \tag{3}$$

For this matrix series to converge, we require $\limsup_{n\to\infty} \|(\mathbf{I} - \frac{1}{V}\mathbf{H}^2)^n\|^{\frac{1}{n}} < 1$. By Gelfand's formula, this limit superior is simply the spectral radius of $\mathbf{I} - \frac{1}{V}\mathbf{H}^2$ which, this being a real symmetric matrix, is exactly the largest of the absolute value of its eigenvalues. Denoting the largest-magnitude eigenvalue of $\mathbf{H}^2$ by $\lambda_{\max}$, our convergence condition is thus equivalent to $V > \frac{1}{2}\lambda_{\max}$. Further, if we strengthen the bound to $V > \lambda_{\max}$, we have that $\left(\mathbf{I} - \frac{1}{V}\mathbf{H}^2\right)^k$ is positive semi-definite for $k = 0, 1, 2, \cdots$, so our series, regardless of where it is truncated, produces a positive semi-definite matrix. We are thus guaranteed to be asymptotically targeting the principal square root. Since $\|\mathbf{H}^2\| \geq \lambda_{\max}$ for any sub-multiplicative norm $\|\cdot\|$, a more practical bound is $V > \|\mathbf{H}^2\|$. See Appendix D for further analysis of the correctness, convergence and behaviour around critical points of this series.

### 3.4 Hessian Products, Choice of $V$ and Series Acceleration

Although we have avoided directly inverting or square-rooting a Hessian-sized matrix, explicitly computing this series remains intractable. Instead, recall that our quantity of interest for second-order optimisation is $\left( \sqrt[+]{\mathbf{H}^2} \right)^{-1} \mathbf{g}$, and consider the series obtained by multiplying (3) by $\mathbf{g}$:

$$(\mathbf{H}^2)^{-\frac{1}{2}}\mathbf{g} = \frac{1}{\sqrt{V}} \sum_{k=0}^{\infty} \frac{1}{2^{2k}} \binom{2k}{k} \left( \mathbf{I} - \frac{1}{V}\mathbf{H}^2 \right)^k \mathbf{g}. \tag{4}$$

Denoting by $\mathbf{a}_k$ the $k$th term of this summation, we have $\mathbf{a}_0 = \frac{1}{\sqrt{V}}\mathbf{g}$ and $\mathbf{a}_k = \frac{2k(2k-1)}{4k^2}\left(\mathbf{a}_{k-1} - \frac{1}{V}\mathbf{H}\mathbf{H}\mathbf{a}_{k-1}\right)$. With two applications of the Hessian-vector product trick (Pearlmutter, 1994), we can compute $\mathbf{H}\mathbf{H}\mathbf{a}_{k-1}$ at the cost of two additional forward and backward passes through the model — a cost vastly smaller than that of storing, manipulating and inverting the full Hessian. By unrolling this recursion, we can thus efficiently compute the summation of a finite number of the $\mathbf{a}_k$.

Under this framework, we have ready access to the product $\mathbf{H}^2\mathbf{g}$, so can use the loose adaptive heuristic $V \geq \frac{\|\mathbf{H}^2\mathbf{g}\|}{\|\mathbf{g}\|}$, which we found to be the most performant strategy for adapting $V$.

In practice, we found (4) to converge slowly, and thus benefit from series acceleration. From a variety of strategies, we found the most successful to be a modification due to Sablonnière (1991) of Wynn's $\epsilon$-algorithm (Wynn, 1956a). Letting $\mathbf{s}_m$ be the $m$th partial sum of (4), the algorithm defines the following recursion:

$$\boldsymbol{\epsilon}_m^{(-1)} = 0, \qquad \boldsymbol{\epsilon}_m^{(0)} = \mathbf{s}_m, \qquad \boldsymbol{\epsilon}_m^{(c)} = \boldsymbol{\epsilon}_{m+1}^{(c-2)} + \left( \left\lfloor \frac{c}{2} \right\rfloor + 1 \right) \left( \boldsymbol{\epsilon}_{m+1}^{(c-1)} - \boldsymbol{\epsilon}_m^{(c-1)} \right)^{-1}. \tag{5}$$

We employ the Samelson vector inverse $\mathbf{a}^{-1} = \frac{\mathbf{a}}{\mathbf{a}^\mathsf{T}\mathbf{a}}$ as suggested by Wynn (1962). Using these definitions, the sequence $\boldsymbol{\epsilon}_m^{(2l)}$ for $m = 0, 1, 2, \cdots$ is the sequence of partial sums of the series $\mathbf{a}_k$ accelerated $l$ times. Thus, we expect the most accurate approximation of (4) to be given by maximising $l$ and $m$, acknowledging there is a corresponding increase in computational cost. Pseudo-code for series acceleration is provided in Appendix A.5.

Algorithm 1 incorporates all these elements to form a complete neural network optimisation algorithm. While expanding the series of (4) to a large number of terms may be arbitrarily expensive, we show in the next Section that useful progress can be made on tractable timescales.

## 4 Experiments

We now move on to empirical evaluation of our algorithm. For all experiments, we use ASHA (Li et al., 2020) to tune each algorithm and dataset combination on the validation loss, sampling 100 random hyperparameter

---

**Algorithm 1** Series of Hessian-Vector Products for Tractable Saddle-Free Newton Optimisation

---

**while** training continues **do**
    Compute training loss, gradient $\mathbf{g}$, Hessian $\mathbf{H}$
    $V \leftarrow \max \left\{ V, \frac{\|\mathbf{H}^2 \mathbf{g}\|}{\|\mathbf{g}\|} \right\}$
    $\mathbf{a}_0, \mathbf{s}_0 \leftarrow \mathbf{g}$
    **for** $k \leftarrow 1$ **to** $K - 1$ **do**
        $\mathbf{a}_k \leftarrow \frac{2k(2k-1)}{4k^2} \left( \mathbf{a}_{k-1} - \frac{1}{V} \mathbf{H}\mathbf{H}\mathbf{a}_{k-1} \right)$
        $\mathbf{s}_k \leftarrow \mathbf{s}_{k-1} + \mathbf{a}_k$
    **end for**
    Compute final term $\hat{\mathbf{s}}_\infty$ after $N$ accelerations of the series $\mathbf{s}_{K-1-2N}, \mathbf{s}_{K-2N}, \cdots, \mathbf{s}_{K-1}$
        (See Algorithm 2 in Appendix A.5)
    $\mathbf{w} \leftarrow \mathbf{w} - \frac{\eta}{\sqrt{V}} \hat{\mathbf{s}}_\infty$
**end while**

---

configurations and setting the maximum available budget based on the model and data combination. Further experimental details and the final hyperparameter settings for all experiments can be found in Appendix A.3, with code available at `https://github.com/rmclarke/SeriesOfHessianVectorProducts`.

We will begin by considering UCI Energy (Tsanas & Xifara, 2012), which is small enough to allow an exact implementation of our algorithm (using eigendecompositions instead of the Neumann series approximation) as a proof of concept, and lends itself to the full-batch setting — the best case scenario for second-order methods. We then move to a setting without these conveniences, namely Fashion-MNIST (Xiao et al., 2017), which is large enough to require require mini-batching and has too many parameters to allow for exact computation of the Hessian. We go on to increasingly difficult scenarios, in terms of both model and dataset size, by considering SVHN (Netzer et al., 2011) and CIFAR-10 (Krizhevsky, 2009) using ResNet-18 architectures.

For UCI Energy, we generate a random dataset split using the same sizes as Gal & Ghahramani (2016); for Fashion-MNIST, SVHN and CIFAR-10 we separate the standard test set and randomly choose $\frac{1}{6}$, $\frac{1}{6}$ and $\frac{1}{10}$ (respectively) of the remaining data to form the validation set. The numerical data for all experiments can be found in Appendix B.1. While we will usually present wall-clock time on the $x$-axis, plots with iteration steps on the $x$-axis are available in Appendix B.2.

For all experiments, we present both training and test loss. The optimisation literature often focuses only on the objective function at hand, i.e. the training loss, since a strong optimiser should be able to solve the function it is given. However, in machine learning our target is always to generalise, i.e. to do well on the unseen test set as a measure of generalisation, and the training loss is only a means toward this end. Since we hope to apply our method to deep learning methods, we consider it important to present both these metrics together.

## 4.1 UCI Energy

We begin with a small-scale experiment on UCI Energy as a proof of concept, training for 6 000 full-batch training epochs. We compare our algorithm to a number of baselines[1]:

**Exact SFN** Full-Hessian implementation of the absolute-value eigenvalue strategy of Pascanu et al. (2014), where we compute the eigenvalue decomposition and take the absolute value of the eigenvalues. We additionally replace eigenvalues near zero with a small constant and then compute the exact inverse of the resulting saddle-free Hessian. For this method, we tune the learning rate, momentum, the threshold for replacing small eigenvalues, and constant which replaces the small eigenvalues.

**Ours** Our implementation of Algorithm 1, using tuned learning rate, momentum, series length $K$ and order of acceleration $N$. As described in Section 3.4, we adapt $V$ using the loose bound $V \geq \frac{\|\mathbf{H}^2 \mathbf{g}\|}{\|\mathbf{g}\|}$ starting

---

[1]We also include an L-BFGS (Liu & Nocedal, 1989) baseline in Appendix B.3

with an initial value of 100, as we found minimal benefit to explicitly tuning $V$. Noting the inherent instability of attempting to invert near-zero eigenvalues, we add a fixed damping coefficient to the Hessian at each use, which we treat as another hyperparameter to optimise. We also considered more accurate approximations to $V$ that would attain a tighter bound (such as computing the largest eigenvalue using power iteration), but found these held little to no benefit.

**SGD** Classical stochastic gradient descent, with a tuned learning rate.

**Adam** (Kingma & Ba, 2015) We tune all the parameters, i.e. learning rate, $\epsilon$, $\beta_1$ and $\beta_2$.

**KFAC (DeepMind)** (Martens & Grosse, 2015) We use the implementation of Botev & Martens (2022) which includes adaptive learning rates, momentum, and damping; we tune the initial damping.

The first algorithm above is an exact version of our algorithm, which is tractable in this particular setting. We also considered including an exact implementation of the Newton second-order update but this diverged rapidly, presumably due to the non-convexity of the optimisation task, so we do not include it here.

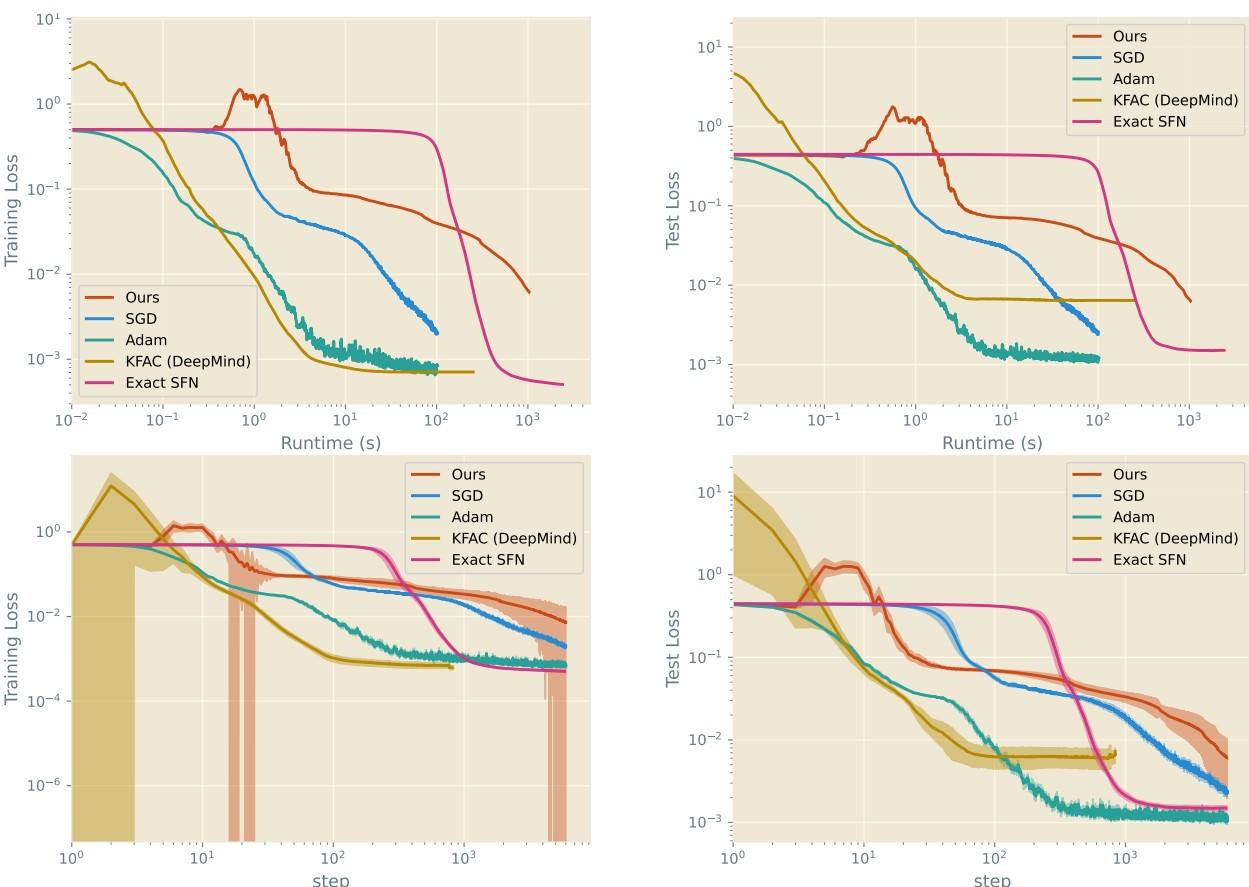

Figure 2: Median training (left) and test (right) MSEs achieved over wall-clock time (top) and training iterations (bottom) on UCI Energy by various optimisers in the full-batch setting, bootstrap-sampled from 50 random seeds. Optimal hyperparameters were tuned with ASHA. Note the logarithmic horizontal axes.

Figure 2 shows the training and test losses both in terms of wall-clock time and as a function of the number of optimisation steps. Exact SFN achieves the best training loss, as we may hope from it being an exact SFN method. This is encouraging, since it provides proof of concept that our approach is generally sensible in the exact setting. However, it does not converge as quickly as may be desired — in comparison, KFAC (DeepMind) and Adam make much faster progress, even when considering the change in loss per iteration, rather than wall-time. Our algorithm deflects from the Exact SFN trend, as its approximate nature would suggest, but does not approach the performance exhibited by KFAC DeepMind.

KFAC DeepMind includes clever adaptation mechanisms and smoothing of the curvature matrix which may give it an advantage over the other algorithms. To investigate this, we include additional variants on the baselines:

**Exact SFN (Adaptive)** Same as Exact SFN, but with adaptive learning rate, momentum and damping strategies as used by KFAC (DeepMind) (see Section 2 and Appendix A.4 for details), as well as an exponential moving average of the curvature matrix. We tune only the initial damping, which subsumes the need for manually replacing small eigenvalues with a constant.

**Ours (Adaptive)** Our implementation of Algorithm 1, incorporating the adaptive learning rate, momentum and damping used by KFAC (DeepMind). We tune the initial damping, number of update steps and order of acceleration.

**KFAC (Kazuki)** (Martens & Grosse, 2015) This corresponds to the default settings for KFAC in Osawa. This version of KFAC is not adaptive and does not smooth the curvature matrix by means of averaging. We tune the damping, learning rate and momentum.

Figure 3 shows the training and test loss profiles in wall-clock time. The best test and training losses are now achieved by Exact SFN (Adaptive). We note that this adaptive version of Exact SFN converges considerably faster than the non-adaptive version, reinforcing our and Martens & Grosse's views on the importance of adapting the learning rate, momentum and damping.

In all cases (KFAC, Exact SFN and Ours), the adaptive version of the algorithm performs significantly better than the non-adaptive version. Although our adaptive algorithm matches Exact SFN and beats SGD and both KFAC versions in terms of final test loss, it is still surpassed by Adam and SFN Exact (Adaptive). Notably, our non-adaptive algorithm does not match the performance of SFN Exact, neither does our adaptive algorithm match SFN Exact (Adaptive). Clearly, we sacrifice training performance by using an approximation to Exact SFN and by not smoothing the curvature matrix.

KFAC (DeepMind) achieves the second best training loss, though not test loss. KFAC (Kazuki) diverges quickly at the start of training, which is also unexpected given that its hyperparameters were tuned and that it behaves reasonably on the later, more difficult problems. We hypothesise that adaptive parameters are an important component of its behaviour and that this setting does not lend itself well to fixed parameters (which is supported by the observation that all the adaptive versions performed better than their non-adaptive counterparts).

In this setting, it seems that short of using exact Hessians, Adam is the best choice of optimiser, displaying the second-best training and test losses and completing faster (in wall-clock time) than the second-order methods. However, we are encouraged that our adaptive algorithm's performance is not far off the exact version and continue to more realistic settings in the sections that follow.

### 4.2 Larger Scale Experiments

Most practical applications are too large to permit full-batch training and so the remainder of our experiments incorporate mini-batching. Since second-order methods may benefit from larger batch sizes, we tune for batch size, choosing from the set $\{50, 100, 200, 400, 800, 1600, 3200\}$.

We show the best (lowest) losses achieved by each algorithm in each problem setting in Figure 5 as well as the training and test loss profiles in Figure 4. Although KFAC (DeepMind) usually attains the best training loss, there is no clear consistent winner in terms of the best test loss achieved across all problems, despite each algorithm having been tuned specifically for each problem.

Surprisingly, KFAC (DeepMind) performs poorly on Fashion-MNIST, where KFAC (Kazuki) and Adam perform well. First-order optimisers seem well-suited to SVHN, where SGD and Adam achieve the best test losses. On CIFAR-10, Adam and the two KFAC variants perform about the same in terms of training loss, but KFAC (DeepMind) performs significantly better in terms of test loss.

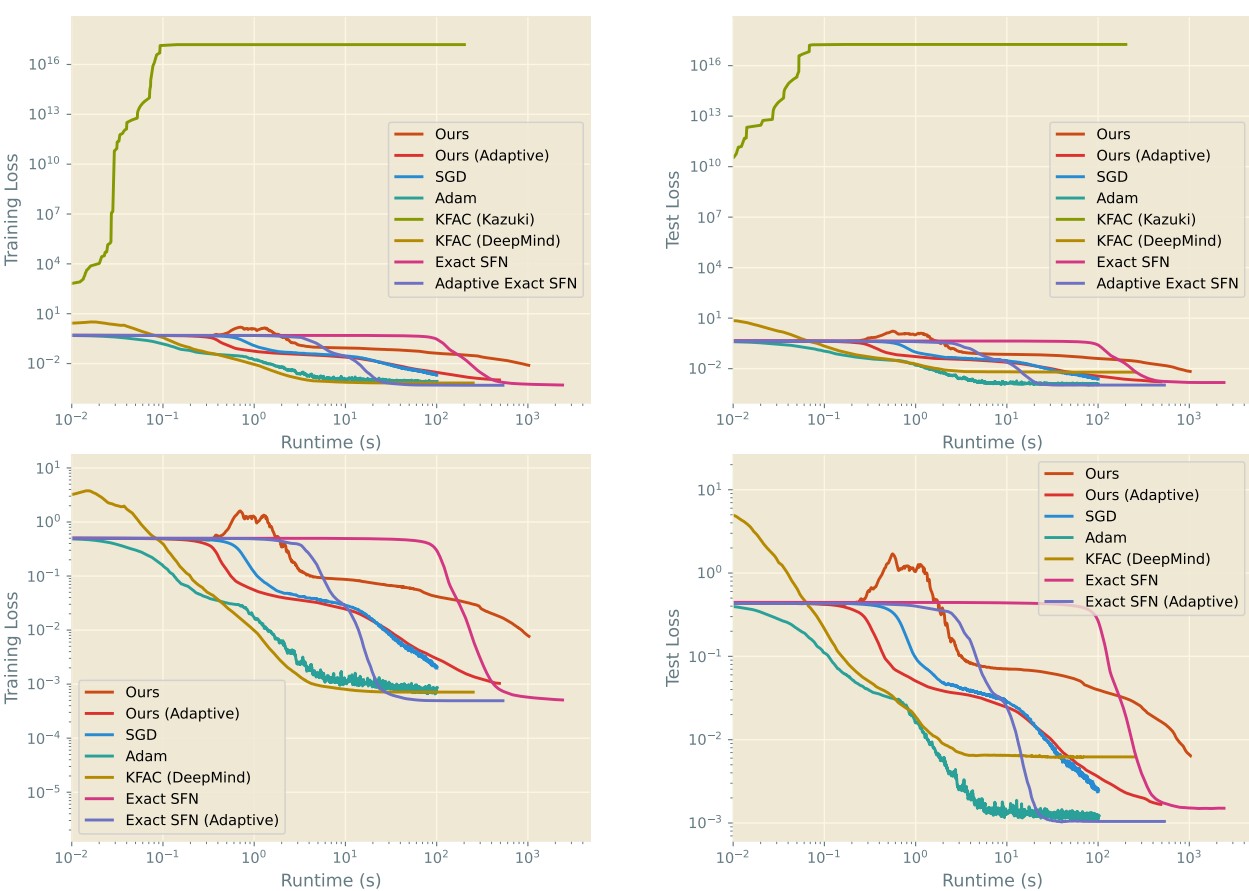

Figure 3: Median training (left) and test (right) MSEs plotted against the log of wall-clock time. The top row includes all additional optimisers; the bottom row excludes KFAC (Kazuki) for clarity. Results are on UCI Energy in the full-batch setting and are bootstrap-sampled from 50 random seeds. Optimal hyperparameters were tuned with ASHA.

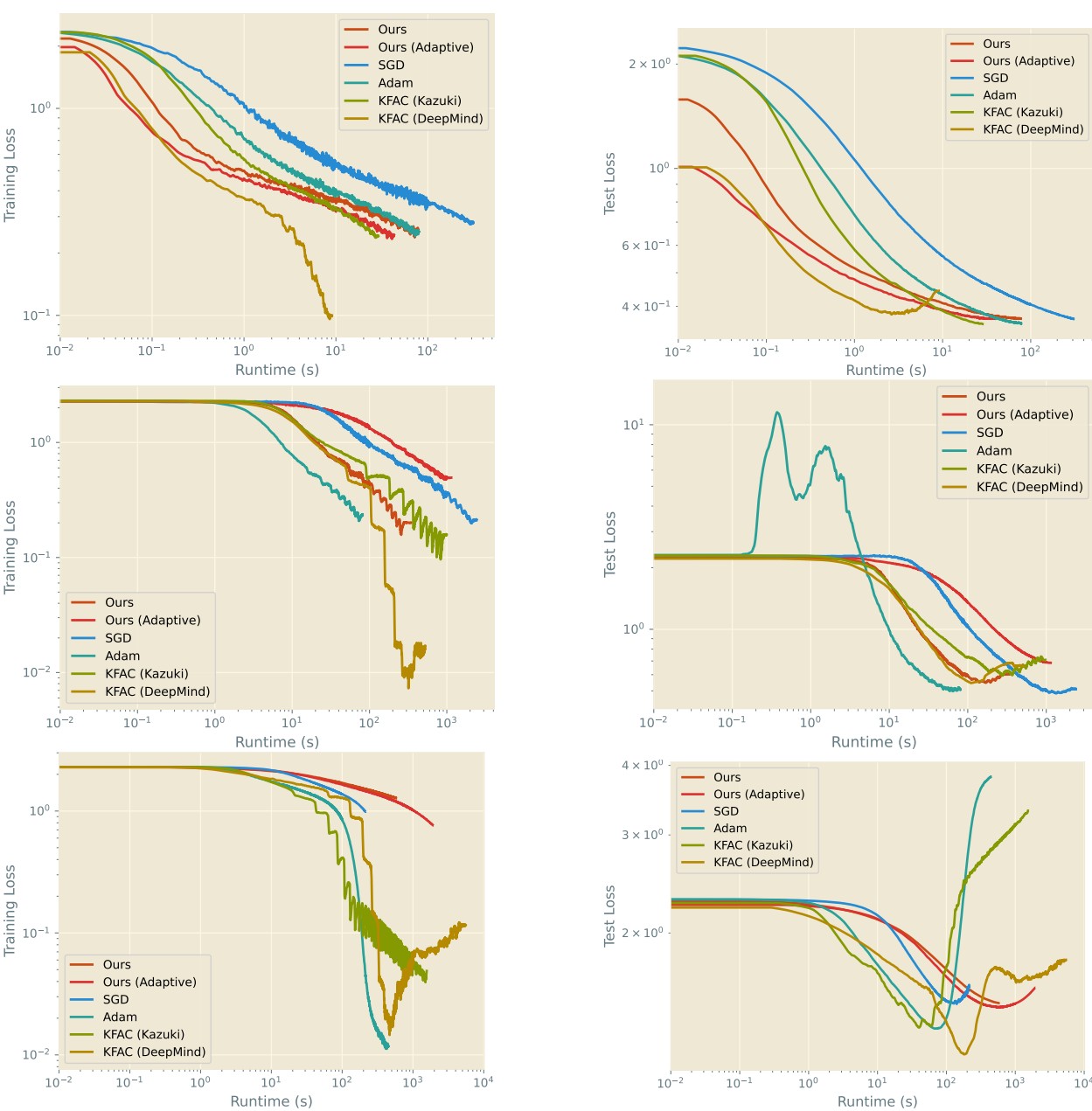

Figure 4: Median training (left) and test (right) loss achieved on Fashion-MNIST (top), SVHN (centre) and CIFAR-10 (bottom) by various optimisers using the optimal hyperparameters chosen by ASHA. Values are bootstrap-sampled from 50 random seeds.

Our findings seem to validate the widespread use of Adam in practice, given its simplicity as compared to KFAC (DeepMind). However, the performance of KFAC (DeepMind) on CIFAR-10 does indicate that there may be benefit to considering second-order optimisers more seriously.

Although our method is not the best on any of the datasets, its performance is not far from that of the other methods. Where the KFAC variants and Adam occasionally diverge during training (see Figure 4), our method is reasonably stable. Moreover, by leveraging large batch sizes, we converge in fewer epochs and less time than the first-order methods in some settings (e.g. Ours (Adaptive) on Fashion-MNIST is faster than both SGD and Adam), despite the additional complexity of our algorithm.

### 4.3 Discussion

We posit that the gap between our performance and expected gains is due to error in our series approximation, of which there are two sources. The first is truncation error, which can be reduced to some extent by increasing the number of terms in the series, though the potency of this will depend on how slow the series is to converge. The second is numerical error: if the Hessian is poorly conditioned, then the repeated multiplications required to compute more terms may cause the series to diverge — even if we have chosen $V$ appropriately, so that the series should converge in theory.

By increasing the number of terms in the series, we can test whether the error is due to truncating the series or numerical error. From our experiment in Appendix B.4 examining the effect of truncation length, we find that for UCI Energy, increasing the number of terms in the series improves performance. However, for the larger-scale problems, we found that increasing the number of steps in the series to be arbitrarily large did not necessarily lead to improved performance. There is thus a trade-off between choosing a sufficiently high number of steps to approximate the desired matrix and choosing sufficiently few to avoid numerical issues. Strategies to improve conditioning of the Hessian may also help to improve this trade-off.

We consider KFAC, which also approximates the curvature, yet proves quite successful on the benchmark suite.[2] KFAC's Kronecker factorisation supports smoothing the curvature estimate with a moving average, which reduces the impact of occasional, poor-quality approximations. Unfortunately, our full-Hessian approximation cannot support such smoothing due to storage requirements. Moreover, KFAC's approximation (which discards the off-diagonal blocks) can be understood intuitively as ignoring the correlations between weights of different layers. In contrast, the rate of convergence of our series varies throughout optimisation, and the impact of truncating the series on the resulting curvature matrix is more difficult to intuit. It may prove fruitful to leverage the same block-diagonal approximation in our method, but with smaller matrices at less risk of ill-conditioning. This would also allow the use of smoothing, which may further improve performance.

There are links between our series approximation and the conjugate gradient (CG) method (Hestenes & Stiefel, 1952). CG solves a linear system of the form $\mathbf{Ax} = \mathbf{b}$ iteratively. At the $k$-th iteration, CG finds the best $\mathbf{x}$ in the $k$-th Krylov subspace (where the $k$-th Krylov subspace is the subspace generated by repeated applications of $\mathbf{A}$ to the residual $\mathbf{r}$, i.e. $\mathcal{K}_k = \mathrm{span}\{\mathbf{r}, \mathbf{Ar}, ..., \mathbf{A}^{k-1}\mathbf{r}\}$) where $\mathbf{r} = \mathbf{b} - \mathbf{Ax}_0$. The inverse Neumann approximation truncated at the $k$-th term also finds a vector in the $k$-th Krylov subspace, but it is not guaranteed to be the optimal one, and so we may expect the Neumann approximation to be worse than CG.[3] Although the series we present in (3) is slightly different, since it is computing the square and square-root at the same time as the inverse, we note that this may provide a clue as to the poor convergence behaviour of the series in general. Future work may consider leveraging insights from the conjugate gradient method to better approximate the inverted saddle-free Hessian.

## 5 Conclusions

In this work, we have motivated, derived and justified an approach to implementing Saddle-Free Newton optimisation of neural networks. By development of an infinite series, we are able to take the absolute

---

[2]In fact, based on KFAC's performance in all our experiments, we found it surprising that KFAC is not more widely utilised in practice. That said, we also found its performance to vary widely between implementations, which may explain this observation.

[3]However, there is literature showing that Neumann series are more stable than CG in neural networks (Shaban et al., 2019; Liao et al., 2018)

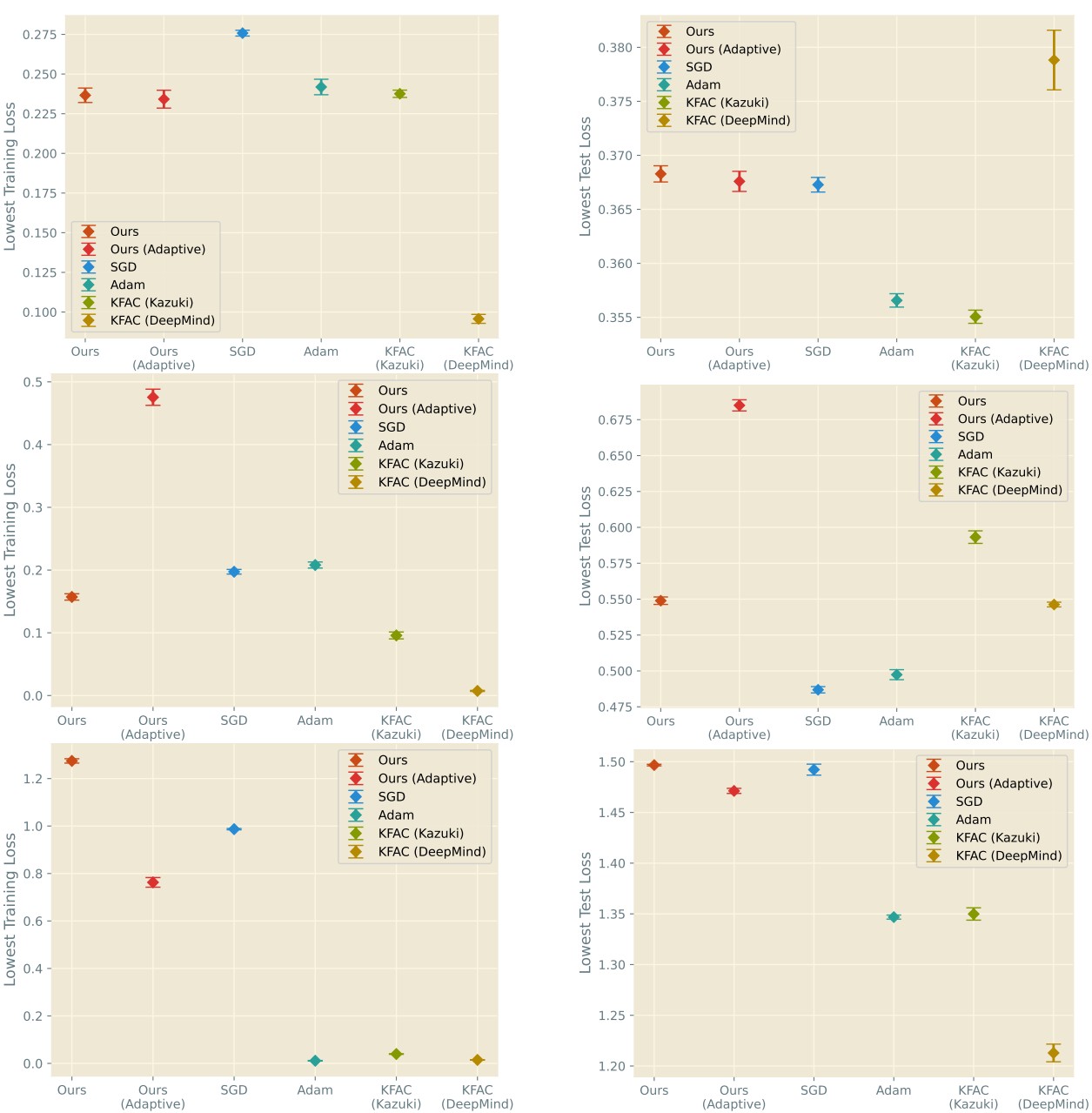

Figure 5: Ranking of optimisers according to lowest training (left) and test (right) losses achieved on Fashion-MNIST (top), SVHN (centre) and CIFAR-10 (bottom). Error bars show standard error in the mean. Values are the minimum of the loss profile across time, generated by bootstrap sampling from 50 random seeds.

values of Hessian eigenvalues without any explicit decomposition. With the additional aid of Hessian-vector products, we further avoid any explicit representation of the Hessian. To our knowledge, this is the first approximate second-order method to take the absolute value of Hessian eigenvalues with an asymptotic exactness guarantee, and whose convergence is limited by compute time rather than available memory. Our algorithm tractably scales to larger networks and datasets, and although it does not consistently outperform Adam or a well-engineered KFAC implementation, its behaviour is comparable to these baselines, in terms of test loss and run time.

Improvements to the inverse approximation such as leveraging Kronecker factorisation or ideas from the conjugate gradient method may provide fruitful avenues of research for future saddle-free Hessian-based optimisation algorithms such as ours. Strategies to reduce numerical error, such as methods to improve the condition number of the Hessian, should also be investigated. Our findings generally support the widespread use of Adam, which performed well on most benchmarks, often beating KFAC despite being a much simpler algorithm. However, the strong performance of KFAC on CIFAR-10, our most complex benchmark, indicates that there may yet be significant gains by applying second-order methods to deep learning.

## Acknowledgements

We are grateful to Richard E. Turner for valuable discussions about this work and its presentation.

We acknowledge computation provided by the CSD3 operated by the University of Cambridge Research Computing Service (`www.csd3.cam.ac.uk`), provided by Dell EMC and Intel using Tier-2 funding from the Engineering and Physical Sciences Research Countil (capital grant EP/P020259/1), and DiRAC funding from the Science and Technology Facilities Council (`www.dirac.ac.uk`).

Ross Clarke acknowledges funding from the Engineering and Physical Sciences Research Council (project reference 2107369, grant EP/S515334/1).

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

## A  Empirical Notes

### A.1  Datasets Used

The datasets we use are all standard in the ML literature; we outline their usage conditions in Table 1.

Table 1: Licences under which we use datasets in this work.

| Dataset | Licence | Source | Input | Output | Total Size |
|---|---|---|---|---|---|
| UCI Energy | Creative Commons Attribution 4.0 International (CC BY 4.0) | Tsanas & Xifara (2012); Gal & Ghahramani (2016) | 8-Vector | Scalar | 692 |
| Fashion-MNIST | MIT | Xiao et al. (2017) | $28 \times 28$ Image | Class (from 10) | 60 000 |
| CIFAR-10 | None specified | Krizhevsky (2009) | $32 \times 32$ Image | Class (from 10) | 60 000 |
| SVHN | None specified | Netzer et al. (2011) | $32 \times 32$ Image | Class (from 10) | 99 289 |

### A.2  Computing Resources Used

The experiments presented were performed using hardware shown in Table 2. All runtime comparisons were thus performed on like-for-like hardware. We make use of GPU acceleration throughout, using the JAX library (Bradbury et al., 2018). Our own code is available at `https://github.com/rmclarke/SeriesOfHessianVectorProducts`.

Table 2: System configurations used to run our experiments.

| Type | CPU | GPU (NVIDIA) | Python | JAX | CUDA |
|---|---|---|---|---|---|
| Cambridge Service for Data Driven Discovery (CSD3)* | AMD EPYC 7763 | Ampere A100 | 3.9.6 | 0.3.25 | 11.1 |

\* `www.csd3.cam.ac.uk`

### A.3  Experimental Hyperparameters

We outline in Tables 3 and 4 the search ranges chosen for our hyperparameter optimisation using ASHA, as well as the best hyperparameters chosen for each setting and the corresponding final losses. Our network architectures and corresponding time budgets are enumerated below:

**UCI Energy** (Tsanas & Xifara, 2012): MLP with 7 hidden layers, each of 12 units (budget 5 minutes). A full training run is 6 000 epochs.

**Fashion-MNIST** (Xiao et al., 2017): MLP with one hidden layer of 50, units (budget 5 minutes). A full training run is 10 epochs.

**SVHN** (Netzer et al., 2011): ResNet-18 (He et al., 2016) (budget 45 minutes). A full training run is 10 epochs.

**CIFAR-10** (Krizhevsky, 2009): ResNet-18 (He et al., 2016) (budget 2 hours). A full training run is 72 epochs.

For KFAC (DeepMind), we set the curvature EMA to 0.95 and did not tune it on advice from the library's author. KFAC allows computational savings by using the same damping parameter and inverse curvature estimate for multiple weight updates, but we set both of these to update at every iteration to match the setting of Ours and Ours (Adaptive). For KFAC Kazuki we set the curvature EMA to zero to turn off the moving average. For KFAC Kazuki, the learning rate, momentum and initial damping were tuned and then fixed (i.e. not adapted).

Since the SFN Exact variants were only applied to UCI Energy, we enumerate those settings here rather than in the tables. For SFN Exact, the optimal settings of tuned parameters were a learning rate of 0.0049 and momentum of 0.0638. The threshold for replacing small eigenvalues was 0.00052 and the replacement constant was $1.54e^{-4}$. For SFN Exact (Adaptive), we only need to tune the initial damping, for which the

Table 3: Details of our hyperparameter search strategy (Part 1). *Ranges* shows the search spaces considered for each hyperparameter as a uniform range, except those marked log, which are sampled from a log-uniform range. Other rows show the optimal hyperparameters chosen to minimise validation loss and the corresponding losses obtained. Note that initial random seeds during tuning were not controlled, so comparisons of the losses achieved by each method must be made with care. All values are rounded to three significant figures.

Ours

| Setting | Batch Size | Learning rate | Momentum | Damping | Number of Series Terms K | Order of Acceleration | Training Loss | Test Loss | Validation Loss |
|---|---|---|---|---|---|---|---|---|---|
| Range | $50 \times 2^{[1,7]}$ | $\log[10^{-3}, 5]$ | $\log[10^{-3}, 0.95]$ | $\log[10^{-8}, 1]$ | $[1, 20]$ | $[0, \frac{K-1}{2}]$ | | | |
| UCI Energy | — | 1.79 | 0.717 | 2.44e-2 | 18 | 8 | 4.55e-4 | 8.62e-4 | 1.29e-3 |
| Fashion-MNIST | 400 | 0.305 | 0.428 | 5.77e-2 | 13 | 5 | 2.15e-1 | 3.72e-1 | 3.38e-1 |
| SVHN | 200 | 0.108 | 0.327 | 1.87e-4 | 1 | 0 | 9.01e-2 | 5.95e-1 | 5.20e-1 |
| CIFAR-10 | 3200 | 0.562 | 0.115 | 2.44e-3 | 2 | 0 | 1.37 | 1.51 | 1.52 |

Ours (Adaptive)

| Setting | Batch Size | Initial Damping | Number of Series Terms (K) | Order of Acceleration | Training Loss | Test Loss | Validation Loss |
|---|---|---|---|---|---|---|---|
| Range | $50 \times 2^{[1,7]}$ | $\log[10^{-8}, 10]$ | $[1, 20]$ | $[0, \frac{K-1}{2}]$ | | | |
| UCI Energy | — | 1.46e-6 | 9 | 4 | 7.15e-3 | 8.17e-3 | 8.45e-3 |
| Fashion-MNIST | 800 | 9.33e-2 | 14 | 2 | 2.45e-1 | 3.65e-1 | 3.35e-1 |
| SVHN | 100 | 4.14e-7 | 5 | 2 | 4.48e-1 | 7.10e-1 | 6.49e-1 |
| CIFAR-10 | 3200 | 1.04e-4 | 5 | 2 | 8.91e-1 | 1.56 | 1.55 |

SGD

| Setting | Batch Size | Learning Rate | Training Loss | Test Loss | Validation Loss |
|---|---|---|---|---|---|
| Range | $50 \times 2^{[1,7]}$ | $\log[10^{-6}, 10^{-1}]$ | | | |
| UCI Energy | — | 9.45e-2 | 7.15e-3 | 8.17e-3 | 8.45e-3 |
| Fashion-MNIST | 50 | 1.88e-2 | 3.81e-1 | 3.69e-1 | 3.42e-1 |
| SVHN | 50 | 1.33e-2 | 2.30e-1 | 4.87e-1 | 4.47e-1 |
| CIFAR-10 | 3200 | 4.11e-3 | 1.01 | 1.57 | 1.56 |

Adam

| Setting | Batch Size | Learning Rate | $\epsilon$ | $1 - \beta_1$ | $1 - \beta_2$ | Training Loss | Test Loss | Validation Loss |
|---|---|---|---|---|---|---|---|---|
| Range | $50 \times 2^{[1,7]}$ | $\log[10^{-6}, 10^0]$ | $\log[10^{-10}, 10^1]$ | $\log[10^{-3}, 10^0]$ | $\log[10^{-4}, 10^0]$ | | | |
| UCI Energy | — | 1.74e-2 | 2.73e-10 | 0.255 | 0.0156 | 3.99e-4 | 8.01e-4 | 7.83e-4 |
| Fashion-MNIST | 200 | 6.55e-4 | 5.99e-5 | 0.475 | 0.0195 | 2.33e-1 | 3.61e-1 | 3.31e-1 |
| SVHN | 1600 | 1.84e-3 | 3.35e-6 | 0.126 | 0.0253 | 1.84e-1 | 5.50e-1 | 5.11e-1 |
| CIFAR-10 | 800 | 7.70e-5 | 3.61e-4 | 0.002 66 | 0.000 100 | 1.40 | 1.44 | 1.45 |

optimal value was 0.0125. We set the decay rate of the exponential moving average to 0.95, as for KFAC (DeepMind).

In the tables below, *Ranges* shows the search spaces considered for each hyperparameter as a uniform range, except those marked log, which are sampled from a log-uniform range. We sampled 100 random configurations for each algorithm and dataset combination. For Adam, we tuned $1 - \beta_1$ and $1 - \beta_2$ using the ranges below and then computed the corresponding values for $\beta_1$ and $\beta_2$. We show the optimal hyperparameters chosen to minimise validation loss and the corresponding losses obtained. Note that initial random seeds during tuning were not controlled, so comparisons of the losses achieved by each method must be made with care. All values are rounded to three significant figures.

### A.4   KFAC Adaptive Heuristics

Here, we give a more technically specific overview of the key adaptive heuristics deployed by KFAC Martens & Grosse (2015), which we presented in Section 2 and employ in our *Adaptive* experimental settings.

Table 4: Details of our hyperparameter search strategy (Part 2), comments as for Table 3

KFAC (DeepMind)

| Setting | Batch Size | Initial Damping | Training Loss | Test Loss | Validation Loss |
|---|---|---|---|---|---|
| Range | $50 \times 2^{[1,7]}$ | $\log[10^{-8}, 10^{0}]$ | | | |
| UCI Energy | — | 6.97e-5 | 2.44e-4 | 1.19e-3 | 8.41e-4 |
| Fashion-MNIST | 3200 | 5.27e-1 | 9.98e-2 | 4.47e-1 | 4.20e-1 |
| SVHN | 1600 | 5.76e-1 | 2.35e-2 | 6.72e-1 | 6.10e-1 |
| CIFAR-10 | 800 | 2.17e-1 | 1.19 | 1.34 | 1.30 |

KFAC (Kazuki)

| Setting | Batch Size | Learning rate | Momentum | Damping | Training Loss | Test Loss | Validation Loss |
|---|---|---|---|---|---|---|---|
| Range | $50 \times 2^{[1,7]}$ | $\log[10^{-6}, 10^{1}]$ | $\log[10^{-3}, 0.95]$ | $\log[10^{-8}, 10^{0}]$ | | | |
| UCI Energy | — | 9.09e-2 | 2.64e-1 | 5.63e-6 | 2.19e-4 | 7.58e-4 | 9.79e-4 |
| Fashion-MNIST | 800 | 8.68e-2 | 4.06e-1 | 2.20e-1 | 2.50e-1 | 3.57e-1 | 3.27e-1 |
| SVHN | 800 | 8.83e-3 | 2.22e-1 | 3.79e-4 | 1.38e-1 | 7.25e-1 | 6.65e-1 |
| CIFAR-10 | 3200 | 2.29e-2 | 6.70e-3 | 2.21e-3 | 9.01e-1 | 1.61 | 1.57 |

**Moving average of curvature matrix** KFAC maintains an online, exponentially-decaying average of the approximate curvature matrix, which improves its approximation thereof and makes the method more robust to stochasticity in mini-batches. For a curvature matrix $\mathbf{C}$ and decay factor $\beta \in (0, 1)$, we have

$$\mathbf{C}_t \leftarrow \beta \mathbf{C}_t + (1 - \beta)\mathbf{C}_{t-1}. \tag{6}$$

**Adaptive learning rate and momentum factor** KFAC's update rule incorporates a learning rate and a momentum factor which are both computed adaptively by assuming a locally quadratic model and solving for the local model's optimal learning rate and momentum factor at every iteration. When the local approximate model has curvature matrix $\mathbf{C}$, gradient $\mathbf{g}$ and our proposed update direction is $\boldsymbol{\Delta}$, we compute the learning rate $\eta$ and momentum $\mu$ by

$$\begin{bmatrix} \eta_t \\ \mu_t \end{bmatrix} = - \begin{bmatrix} \boldsymbol{\Delta}_t^\mathsf{T} \mathbf{C} \boldsymbol{\Delta}_t & \boldsymbol{\Delta}_t^\mathsf{T} \mathbf{C}(\mathbf{x}_t - \mathbf{x}_{t-1}) \\ \boldsymbol{\Delta}_t^\mathsf{T} \mathbf{C}(\mathbf{x}_t - \mathbf{x}_{t-1}) & (\mathbf{x}_t - \mathbf{x}_{t-1})^\mathsf{T} \mathbf{C}(\mathbf{x}_t - \mathbf{x}_{t-1}) \end{bmatrix}^{-1} \begin{bmatrix} \mathbf{g}_t^\mathsf{T} \boldsymbol{\Delta}_t \\ \mathbf{g}_t^\mathsf{T}(\mathbf{x}_t - \mathbf{x}_{t-1}) \end{bmatrix} \tag{7}$$

.

**Tikhonov damping with Levenberg-Marquardt style adaptation.** KFAC incorporates two damping terms: $\eta$ for weight regularisation, and $\lambda$ which is adapted throughout training using Levenberg-Marquardt style updates (Moré, 1978). The damping constant $\lambda$ can be interpreted as defining a trust region for the update step. When the curvature matrix matches the observed landscape, the trust region is grown by shrinking $\lambda$ and vice versa. This level of "mismatch" is captured by the ratio of the actual change in loss to the change predicted by the locally quadratic model. If the ratio is near one and thus the local quadratic model matches the observed losses well, then the curvature matrix is a useful approximation to the local landscape and damping is decreased (i.e. the trust region is increased). Conversely, if the ratio is far from one (implying the local model is not accurate), the damping is increased so that optimisation becomes more SGD-like (i.e. the trust region is reduced).

In notation, if the objective function is $f(\mathbf{x})$, $\widehat{f}(\mathbf{x})$ is our local quadratic estimate of $f(\mathbf{x})$ and we have some adjustment factor $\omega \in (0, 1)$, KFAC updates the damping $\lambda$ by the following rule:

$$\rho_t = \frac{f(\mathbf{x}_t) - f(\mathbf{x}_{t-1})}{\widehat{f}(\mathbf{x}_t) - \widehat{f}(\mathbf{x}_{t-1})} \qquad \lambda_{t+1} = \begin{cases} \frac{1}{\omega}\lambda_t & \rho_t < \frac{1}{4} \\ \omega\lambda_t & \rho_t > \frac{3}{4} \\ \lambda_t & \text{otherwise} \end{cases} . \tag{8}$$

### A.5 Series Acceleration

Recall the Sablonnière (1991)-accelerated Wynn $\epsilon$-algorithm (Wynn, 1956a) applied to the series of $m$th partial sums $\mathbf{s}_m$ gives the recursion

$$\epsilon_m^{(-1)} = 0, \qquad \epsilon_m^{(0)} = \mathbf{s}_m, \qquad \epsilon_m^{(c)} = \epsilon_{m+1}^{(c-2)} + \left( \left\lfloor \frac{c}{2} \right\rfloor + 1 \right) \left( \epsilon_{m+1}^{(c-1)} - \epsilon_m^{(c-1)} \right)^{-1}. \tag{9}$$

This definition is sufficient to compute the accelerated series, but a naïve implementation requires all the $\epsilon$ terms to be stored in memory, which rapidly becomes problematic for larger networks. By carefully defining the order in which these terms are computed, we may substantially reduce the intermediate memory storage required. Such a strategy was outlined by Wynn (1962), but a combination of changing conventions and unclear formatting make it difficult to interpret; we present our own derivation of the same process in Algorithm 2.

---
**Algorithm 2** Sablonnière-Modified Wynn $\epsilon$-Algorithm with Samelson Inverse
---
**Require:** Sequence $\mathbf{p}_0, \mathbf{p}_1, \cdots, \mathbf{p}_{2N}$ and acceleration order $N$
   **for** $m \leftarrow 0$ **to** $2N$ **do**
      $\epsilon_m^{(0)} \leftarrow \mathbf{p}_m, \quad \epsilon_{m+1}^{(-1)} \leftarrow \mathbf{0}$
   **end for**
   $m \leftarrow 0, \quad c \leftarrow 1$
   **while** $m \leq 2N$ **do**
      **while** $m \geq 0$ **do**
         $\epsilon_m^{(c)} = \epsilon_{m+1}^{(c+2)} + \left( \lfloor \frac{c}{2} \rfloor + 1 \right) \left( \epsilon_{m+1}^{(c-1)} - \epsilon_m^{(c-1)} \right)^{-1}$
         Delete $\epsilon_{m+1}^{(c-2)}$
         $m \leftarrow m - 1, \quad c \leftarrow c + 1$
      **end while**
      Delete $\epsilon_{m+1}^{(c-2)}$
      $m \leftarrow c - 1, \quad c \leftarrow 1$
   **end while**
   **return** $\epsilon_0^{(2N)}$
---

Table 5: We provide the median of the training loss, test loss and validation loss for bootstrap-samples from 50 random seeds. Medians values are rounded to four significant figures; errors are rounded to two significant figures.

| Setting | Algorithm | Training Loss | Test Loss | Validation Loss |
|---|---|---|---|---|
| UCI Energy | Ours | $0.006459 \pm 0.005$ | $0.007087 \pm 0.0066$ | $0.0104 \pm 0.0083$ |
| | Ours (Adaptive) | $0.001029 \pm 4\text{e-}05$ | $0.001678 \pm 9.5\text{e-}05$ | $0.002165 \pm 0.00012$ |
| | SGD | $0.001947 \pm 0.0002$ | $0.002361 \pm 0.00025$ | $0.003191 \pm 0.00038$ |
| | Adam | $0.000657 \pm 4\text{e-}05$ | $0.00113 \pm 8.9\text{e-}05$ | $0.001571 \pm 0.00011$ |
| | KFAC (Kazuki) | $0.5018 \pm 0.00056$ | $881.8 \pm 6.2\text{e+}02$ | $904.1 \pm 8.1\text{e+}02$ |
| | KFAC (DeepMind) | $0.000714 \pm 0.00012$ | $0.006663 \pm 0.0024$ | $0.008849 \pm 0.0031$ |
| | Exact SFN | $0.0005019 \pm 2.4\text{e-}05$ | $0.0015 \pm 9.2\text{e-}05$ | $0.001994 \pm 9.7\text{e-}05$ |
| | Exact SFN (Adaptive) | $0.0004941 \pm 1.4\text{e-}05$ | $0.001045 \pm 3.7\text{e-}05$ | $0.001407 \pm 3.4\text{e-}05$ |
| | LBFGS | $0.002619 \pm 0.00051$ | $0.002929 \pm 0.00081$ | $0.004161 \pm 0.00069$ |
| Fashion-MNIST | Ours | $0.237 \pm 0.0061$ | $0.3691 \pm 0.00081$ | $0.3423 \pm 0.00065$ |
| | Ours (Adaptive) | $0.233 \pm 0.0054$ | $0.3684 \pm 0.0009$ | $0.3429 \pm 0.0011$ |
| | SGD | $0.2762 \pm 0.0024$ | $0.3693 \pm 0.00062$ | $0.3436 \pm 0.00053$ |
| | Adam | $0.2425 \pm 0.0054$ | $0.3582 \pm 0.00086$ | $0.3312 \pm 0.00036$ |
| | KFAC (Kazuki) | $0.2375 \pm 0.0019$ | $0.3566 \pm 0.0006$ | $0.3328 \pm 0.00078$ |
| | KFAC (DeepMind) | $0.09567 \pm 0.0027$ | $0.4423 \pm 0.0023$ | $0.4222 \pm 0.0022$ |
| SVHN | Ours | $0.1573 \pm 0.0049$ | $0.583 \pm 0.0033$ | $0.5205 \pm 0.0021$ |
| | Ours (Adaptive) | $0.4757 \pm 0.014$ | $0.6901 \pm 0.0041$ | $0.6335 \pm 0.0036$ |
| | SGD | $0.1979 \pm 0.0036$ | $0.5113 \pm 0.0024$ | $0.4503 \pm 0.0024$ |
| | Adam | $0.2082 \pm 0.0042$ | $0.5029 \pm 0.0045$ | $0.4514 \pm 0.004$ |
| | KFAC (Kazuki) | $0.09535 \pm 0.006$ | $0.7036 \pm 0.0047$ | $0.6288 \pm 0.0048$ |
| | KFAC (DeepMind) | $0.007294 \pm 0.00078$ | $0.6854 \pm 0.0026$ | $0.6288 \pm 0.0035$ |
| CIFAR-10 | Ours | $1.275 \pm 0.009$ | $1.497 \pm 0.00098$ | $1.504 \pm 0.0019$ |
| | Ours (Adaptive) | $0.7614 \pm 0.02$ | $1.595 \pm 0.0084$ | $1.596 \pm 0.0076$ |
| | SGD | $0.9871 \pm 0.0027$ | $1.597 \pm 0.017$ | $1.617 \pm 0.014$ |
| | Adam | $0.01125 \pm 0.0012$ | $3.803 \pm 0.0091$ | $3.761 \pm 0.012$ |
| | KFAC (Kazuki) | $0.03947 \pm 0.0011$ | $3.29 \pm 0.0075$ | $3.226 \pm 0.013$ |
| | KFAC (DeepMind) | $0.01466 \pm 0.0012$ | $1.727 \pm 0.019$ | $1.683 \pm 0.019$ |

## B  Additional Results

### B.1  Tabulated Results

We present the results from Section 4 in tabular form in Table 5.

### B.2  Test and Training Trajectories Per Iteration

In Figures 6 and 7, we present complementary plots for the experiments in Section 4, showing the median training and test losses plotted as a function of weight update steps rather than time.

### B.3  L-BFGS Baseline

In this section, we include plots for UCI Energy with L-BFGS (Liu & Nocedal, 1989) included as an additional baseline. Numerical results for L-BFGS are included in Table 5 as well.

We used the version of L-BFGS in the JAX library (version 0.3.14). We set the number of optimisation steps in the main loop to 20, the maximum number of function evaluations to 25 and the maximum number of Jacobian evaluations to 100. Since larger values are almost always better for all these parameters, we set them to the largest values within our hardware constraints that allowed for a comparable runtime.

### B.4  Effect of Truncation Length

In practice, we wish to avoid computing many terms from the series approximation to the inverse saddle-free Hessian in Equation (4). We may then ask: how many terms are sufficient? Here, we investigate that question

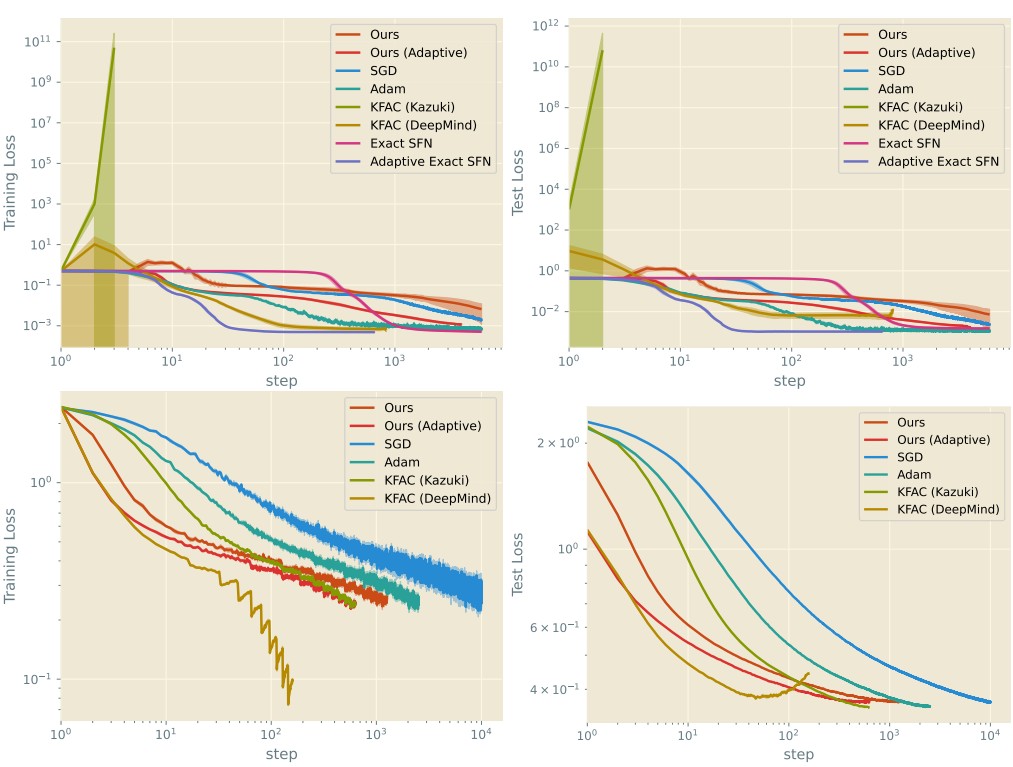

Figure 6: Median training (left) and test (right) loss achieved on UCI Energy (top) and Fashion-MNIST (bottom) plotted per iteration of training. Values are bootstrap-sampled from 50 random seeds. Optimal hyperparameters were tuned with ASHA.

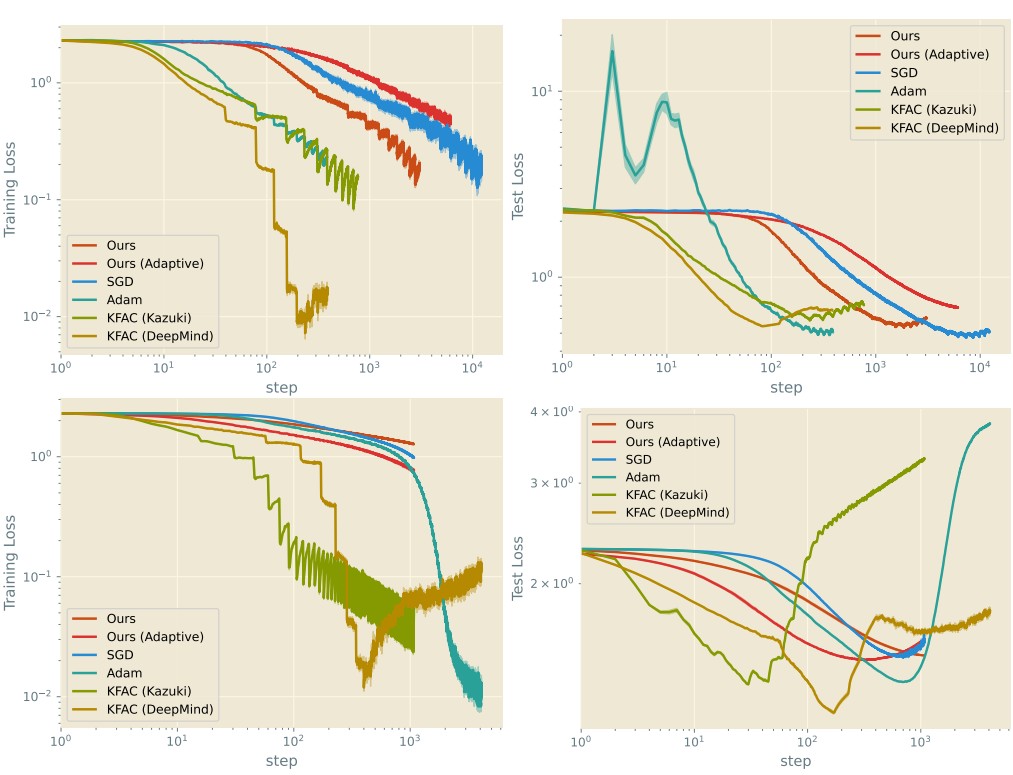

Figure 7: Median training (left) and test (right) loss achieved on SVHN (top) and CIFAR-10 (bottom) plotted per iteration of training. Values are bootstrap-sampled from 50 random seeds. Optimal hyperparameters were tuned with ASHA.

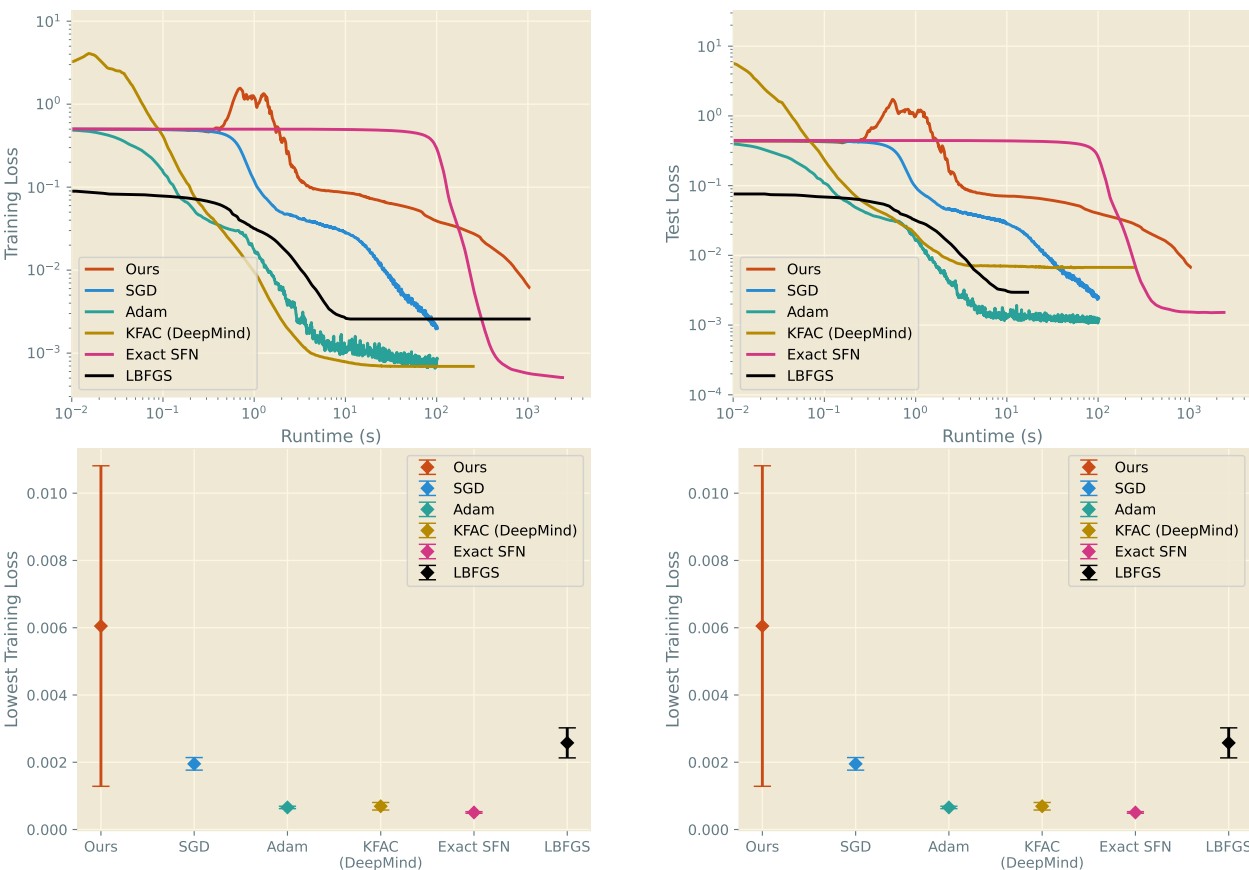

Figure 8: Median training (left) and test (right) MSEs achieved on UCI Energy by various optimisers including L-BFGS in the full-batch setting, bootstrap-sampled from 50 random seeds. Optimal hyperparameters were tuned with ASHA. The x-axes are wall-clock time in the top row, and training iteration in the bottom row; note the log-scaling in both cases.

empirically by applying our method to UCI Energy, but varying the number of terms, $K$ used to approximate the series. As shown in Figure 9, we see clear improvement as the number of computed terms is increased, but even computing only three terms provides a sufficiently close approximation to the saddle-free Hessian for us to reach reasonable loss values. We consider further theoretical justification for this in Appendix D.2.

## B.5    Comparison of Series Accelerators

While developing our algorithm, we considered a range of series accelerators:

- Shanks transformation (Schmidt, 1941; Shanks, 1955), which is implemented by Wynn's $\epsilon$-algorithm (Wynn, 1956a)
- Sablonnière (1991) modification of the Wynn $\epsilon$-algorithm
- Levin-$t$ transform (Levin, 1972), which we found more stable than the related $u$ and $v$ transforms
- Padé approximants (Graves-Morris, 1994)

We also investigated the vector- and topological-$\rho$ accelerators (Wynn, 1956b; Osada, 1991) and the $d^{(2)}$ transformation (Levin & Sidi, 1981; Osada, 1996), but found these to be markedly less robust, so do not show results here.

To investigate the relative merits of these accelerators, we randomly populate a 100-dimensional Hessian matrix $\mathbf{H}$ with independent draws from a standard normal distribution, from which we compute the exact

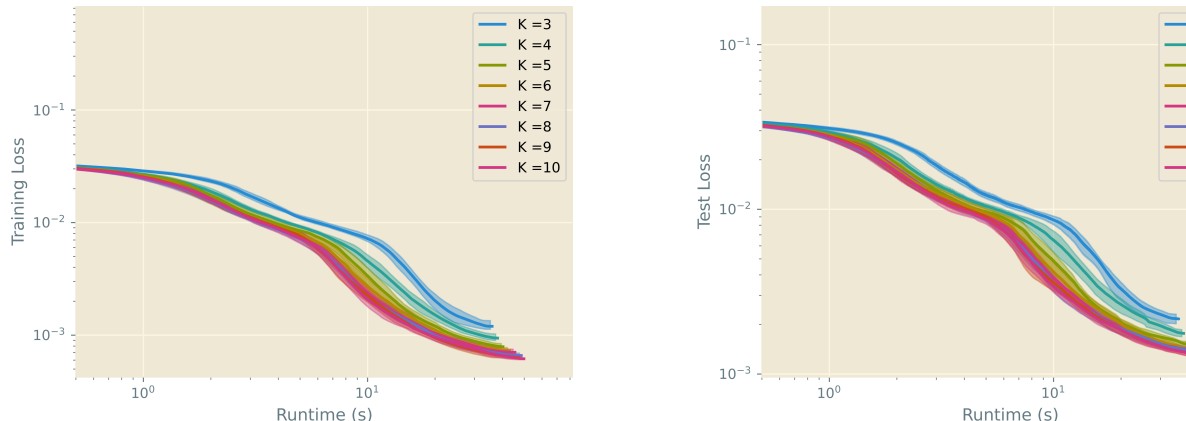

Figure 9: We consider the effect of varying the number of series terms $K$ used to approximate the saddle-free Hessian in (4). The number of steps varies from three to ten. All other settings are as in Section 4.1, including the acceleration order $N = 1$. The results above are bootstrap sampled from 50 different random seeds.

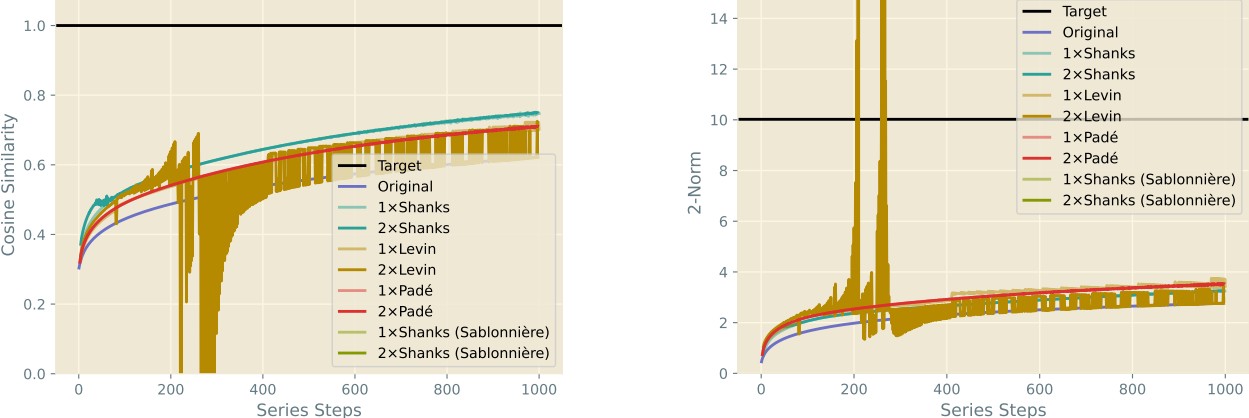

Figure 10: Comparison of series acceleration techniques. From a randomly-initialised Hessian, we compute an exact saddle-free update vector (—), then compare its direction (by cosine similarity; left) and magnitude (by 2-norm; right) with those vectors found by our unmodified series (—) and a variety of accelerators applied to that series, as the latter vary when progressively more series terms are considered.

vector $\left(\sqrt[+]{\mathbf{H}^2}\right)^{-1}\mathbf{g}$ at some random starting point. With this exact target in mind, we compute $1\,000$ steps of our approximating series, then apply each acceleration algorithm in turn up to four times, comparing the resulting directional error (by cosine magnitude) and magnitude error (by 2-norm) of our update step. Since the differences between accelerators dwarfed those between different acceleration orders of the same accelerator, we show only the acceleration orders $N = 1, 2$ in our results (Figure 10). Note that Shanks acceleration and its Sablonnière modification are indistinguishable at the scale of these plots.

From these plots, we observe Shanks acceleration, and its Sablonnière modification, to reliably converge faster towards the correct update direction than the other accelerators, although Padé acceleration marginally beats these when we compare update magnitudes. Every accelerator makes progress faster than the original series, though Levin-$t$ acceleration seems insufficiently robust for our purposes.

As no accelerator comes particularly close to our target vector, there clearly remains some improvement to be made at managing this series' convergence. Noting that the greatest acceleration benefit is seen for early series steps, we choose to focus on this window. Further, since the update magnitude is generally *under*estimated, we prioritise the update direction through the cosine similarity metric, as we expect too-short steps in the correct direction to retain stable optimisation behaviour. Subjectively, Sablonnière's modification of Shanks' algorithm was slightly more stable in our experiments, so we select this accelerator to use in this paper.

## C    Detailed Derivations

In this Section, we provide a more verbose derivation of the key results of Section 3.

### C.1    Scalar Inverse Square-Root Series

The generalised binomial theorem provides a means of writing the quantity $(x + y)^r$ as the infinite series

$$(x + y)^r = \sum_{k=0}^{\infty} \binom{r}{k} x^{r-k} y^k, \tag{10}$$

where the generalisation admits any complex $r$ using the definition

$$\binom{r}{k} = \frac{r(r-1)(r-2)\cdots(r-k+1)}{k!}. \tag{11}$$

In particular, we have

$$(1-z)^{-\frac{1}{2}} = \sum_{k=0}^{\infty} \binom{-\frac{1}{2}}{k}(-z)^k \tag{12}$$

$$= \sum_{k=0}^{\infty} \frac{(-\frac{1}{2})(-\frac{1}{2}-1)(-\frac{1}{2}-2)\cdots(-\frac{1}{2}-k+1)}{k!}(-1)^k z^k \tag{13}$$

$$= \sum_{k=0}^{\infty} \frac{(\frac{1}{2})(\frac{1}{2}+1)(\frac{1}{2}+2)\cdots(\frac{1}{2}+k-1)}{k!}(-1)^{2k} z^k \tag{14}$$

$$= \sum_{k=0}^{\infty} \frac{(\frac{1}{2})(\frac{1}{2}+1)(\frac{1}{2}+2)\cdots(\frac{1}{2}+k-1)}{k!} z^k \tag{15}$$

$$= \sum_{k=0}^{\infty} \frac{(1)(1+2)(1+4)\cdots(2k-1)}{k!} \frac{1}{2^k} z^k \tag{16}$$

$$= \sum_{k=0}^{\infty} \frac{(2k-1)!}{k!(2k-2)(2k-4)(2k-6)\cdots(2)} \frac{1}{2^k} z^k \tag{17}$$

$$= \sum_{k=0}^{\infty} \frac{(2k-1)!}{k!(k-1)(k-2)(k-3)\cdots(1)} \frac{1}{2^{k-1} 2^k} z^k \tag{18}$$

$$= \sum_{k=0}^{\infty} \frac{(2k-1)!}{k!(k-1)!} \frac{1}{2^{2k-1}} z^k \tag{19}$$

$$= \sum_{k=0}^{\infty} \frac{(2k)!}{k!k!} \frac{k}{2k} \frac{1}{2^{2k-1}} z^k \tag{20}$$

$$= \sum_{k=0}^{\infty} \frac{1}{2^{2k}} \binom{2k}{k} z^k \tag{21}$$

### C.2    Series Convergence

Denote by $a_k$ the $k$th term of the summation of (21). For this series to be convergent, it suffices that

$$\limsup_{n \to \infty} |a_n|^{\frac{1}{n}} < 1 \tag{22}$$

by the root test. Applying this test to our series yields

$$\limsup_{n\to\infty} |a_n|^{\frac{1}{n}} = \limsup_{n\to\infty} \left| \frac{1}{2^{2n}} \binom{2n}{n} z^n \right|^{\frac{1}{n}} \tag{23}$$

$$= \limsup_{n\to\infty} \frac{1}{2^2} \left( \frac{(2n)!}{n!n!} \right)^{\frac{1}{n}} |z^n|^{\frac{1}{n}} \tag{24}$$

$$= \frac{1}{4} \limsup_{n\to\infty} \left( \frac{(2n)!}{n!n!} \right)^{\frac{1}{n}} |z^n|^{\frac{1}{n}} \tag{25}$$

$$\leq \frac{1}{4} \limsup_{n\to\infty} \left( \frac{(2n)^{2n}}{n^{2n}} \right)^{\frac{1}{n}} |z^n|^{\frac{1}{n}} \tag{26}$$

$$= \frac{1}{4} \limsup_{n\to\infty} \frac{4n^2}{n^2} |z^n|^{\frac{1}{n}} \tag{27}$$

$$= \limsup_{n\to\infty} |z^n|^{\frac{1}{n}} < 1 \tag{28}$$

Thus, for the series to converge, it is sufficient that $\limsup_{n\to\infty} |z^n|^{\frac{1}{n}} < 1$.

### C.3 Matrix Extension and Scaling

This series extends naturally to the matrix case by choosing a square matrix to substitute for $z$ and replacing 1 with the appropriately-sized identity matrix $\mathbf{I}$. Ideally, we would choose $z = \mathbf{I} - \mathbf{H}^2$ and immediately recover a series expression for $\left(\mathbf{H}^2\right)^{-\frac{1}{2}}$. However, as we will observe, such a $z$ will not allow the series to converge for arbitrary $\mathbf{H}$, so we will instead introduce a scaling factor $V$ and write $z = \mathbf{I} - \frac{1}{V}\mathbf{H}^2$.

The scalar convergence condition $\limsup_{n\to\infty} |z^n|^{\frac{1}{n}} < 1$ generalises naturally to the matrix case. Let $\|\cdot\|$ denote any compatible sub-multiplicative matrix norm — that is, one which satisfies $\|\mathbf{A}\mathbf{B}\| \leq \|\mathbf{A}\| \|\mathbf{B}\|$ and $\|\mathbf{A}\mathbf{x}\| \leq \|\mathbf{A}\| \|\mathbf{x}\|$ for all dimensionally-compatible matrices $\mathbf{A}, \mathbf{B}$ and vectors $\mathbf{x}$. This definition includes all matrix norms induced by vector norms. Then, the convergence condition becomes $\limsup_{n\to\infty} \|z^n\|^{\frac{1}{n}} < 1$.

Collecting these extensions, we recover the series

$$\left(\mathbf{H}^2\right)^{-\frac{1}{2}} = V^{-\frac{1}{2}} \sum_{k=0}^{\infty} \frac{1}{2^{2k}} \binom{2k}{k} \left( \mathbf{I} - \frac{1}{V}\mathbf{H}^2 \right)^k \tag{29}$$

and the convergence condition

$$\limsup_{n\to\infty} \|z^n\|^{\frac{1}{n}} = \limsup_{n\to\infty} \left\| \left( \mathbf{I} - \frac{1}{V}\mathbf{H}^2 \right)^n \right\|^{\frac{1}{n}} < 1. \tag{30}$$

Gelfand's formula gives that, for any matrix norm, $\limsup_{n\to\infty} \left\| \left( \mathbf{I} - \frac{1}{V}\mathbf{H}^2 \right)^n \right\|^{\frac{1}{n}}$ is equal to the spectral radius of $\mathbf{I} - \frac{1}{V}\mathbf{H}^2$. Since we are working with real, symmetric matrices, their eigenvalues are all real, whence the spectral radius is simply the largest of the absolute values of the eigenvalues of $\mathbf{I} - \frac{1}{V}\mathbf{H}^2$.

Let $\lambda$ be an arbitrary eigenvalue of $\mathbf{H}^2$. By reference to the eigendecomposition of $\mathbf{H}^2$, the corresponding eigenvalue of $\mathbf{I} - \frac{1}{V}\mathbf{H}^2$ is $1 - \frac{1}{V}\lambda$. Thus, for the spectral radius of $\mathbf{I} - \frac{1}{V}\mathbf{H}^2$ to be less than unity, we require for all eigenvalues $\lambda$ of $\mathbf{H}^2$ that

$$-1 < 1 - \frac{1}{V}\lambda < 1 \tag{31}$$

$$\implies 0 < \lambda < 2V. \tag{32}$$

Now, since $\mathbf{H}^2$ is positive semi-definite by construction, we have $\lambda \geq 0$, and our implicit assumption of the invertibility of $\mathbf{H}$ (and hence $\mathbf{H}^2$) gives $\lambda \neq 0$, whence we recover $\lambda > 0$ as required. For the upper bound, it

suffices to consider only the largest eigenvalue of $\mathbf{H}^2$, which we denote by $\lambda_{\max}$. We thus secure convergence by the condition

$$V > \frac{1}{2}\lambda_{\max}. \tag{33}$$

We would prefer to compute this bound on $V$ without explicit reference to the largest eigenvalue of $\mathbf{H}^2$, which may be expensive to compute in general. Instead, let $\mathbf{u}_{\max}$ be the corresponding eigenvector of $\lambda_{\max}$. Then, by sub-multiplicativity of the matrix norm, we have

$$\left\|\mathbf{H}^2\right\| \left\|\mathbf{u}_{\max}\right\| \geq \left\|\mathbf{H}^2\mathbf{u}_{\max}\right\| \tag{34}$$
$$= \left\|\lambda_{\max}\mathbf{u}_{\max}\right\| \tag{35}$$
$$= \lambda_{\max}\left\|\mathbf{u}_{\max}\right\| \tag{36}$$
$$\implies \left\|\mathbf{H}^2\right\| \geq \lambda_{\max}. \tag{37}$$

So for convergence of the series, it is sufficient that, for any sub-multiplicative norm $\|\cdot\|$:

$$V > \frac{1}{2}\left\|\mathbf{H}^2\right\|. \tag{38}$$

### C.4 Principality of Square Root

Recall that the principal square root of $\mathbf{H}^2$ is positive semi-definite by construction. The inverse of the principal square root, where it exists, must also then be positive semi-definite. So if the result of our series is positive semi-definite, it must have computed the principal square root.

Consider again our series from (29):

$$(\mathbf{H}^2)^{-\frac{1}{2}} = V^{-\frac{1}{2}} \sum_{k=0}^{\infty} \frac{1}{2^{2k}} \binom{2k}{k} \left(\mathbf{I} - \frac{1}{V}\mathbf{H}^2\right)^k. \tag{39}$$

Under our convergence condition $V > \frac{1}{2}\lambda_{\max} \leq \frac{1}{2}\left\|\mathbf{H}^2\right\|$, we have that the eigenvalues of $\mathbf{I} - \frac{1}{V}\mathbf{H}^2$ all fall within $(-1, 1)$. However, if we strengthen the bound on $V$ to $V > \lambda_{\max}$, reprising the argument of the previous section gives that the eigenvalues of $\mathbf{I} - \frac{1}{V}\mathbf{H}^2$ must fall within $[0, 1]$, making $\mathbf{I} - \frac{1}{V}\mathbf{H}^2$ positive semi-definite. But then $\left(\mathbf{I} - \frac{1}{V}\mathbf{H}^2\right)^k$ is also positive semi-definite for $k = 0, 1, 2, \cdots$. This means our series is a linear combination of positive semi-definite matrices with positive coefficients, so the summation — even when truncated to a finite number of terms — must be positive semi-definite. Thus, our construction has computed the inverse of the principal square root of $\mathbf{H}^2$, as required, when we use the stronger condition

$$V > \lambda_{\max} \tag{40}$$
$$\Longleftarrow V > \left\|\mathbf{H}^2\right\|. \tag{41}$$

# D  Algorithm Analysis

## D.1  Choice of Scaling Factor

Since $V$ exists only to suitably scale the matrix $\mathbf{H}^2$, we have some freedom in its choice of value. We hypothesise that a smaller $V$ (representing a tighter fit of our convergence bound) would best mitigate any issues with numerical precision, as this avoids rescaling values more than necessary. Although we also hypothesise, based on results for the scalar series, that a larger $V$ would ensure more rapid convergence of the series, our subsequent rescaling outside the summation most likely eliminates any gains here. Thus, we seek a $V$ which satisfies our bound $V > \lambda_{\max} \leq \|\mathbf{H}^2\|$ as tightly as possible, but which may be calculated without excessive computational cost.

A naïve approach is to note that $\mathrm{tr}(\mathbf{H}^2)$ is the sum of the (guaranteed non-negative) eigenvalues of $\mathbf{H}^2$, so is certainly an upper bound on the largest. Denoting the dimensionality of $\mathbf{H}$ by $m \times m$, we then have

$$\mathrm{tr}(\mathbf{H}^2)\,\mathrm{tr}\left(\mathbf{I}^2\right) \leq (\mathrm{tr}\,\mathbf{H})^2 (\mathrm{tr}\,\mathbf{I})^2 \tag{42}$$

$$m\,\mathrm{tr}(\mathbf{H}^2) \leq m^2 (\mathrm{tr}\,\mathbf{H})^2 \tag{43}$$

$$\implies \mathrm{tr}(\mathbf{H}^2) \leq m(\mathrm{tr}\,\mathbf{H})^2, \tag{44}$$

so it suffices to set $V = m(\mathrm{tr}\,\mathbf{H})^2$. Since the diagonal elements of $\mathbf{H}$ are the unmixed second derivatives, we can compute them efficiently by differentiating every element of the gradient vector $\mathbf{g}$ with respect to its corresponding weight parameter, and thus compute $\mathrm{tr}\,\mathbf{H}$ without explicitly computing $\mathbf{H}$. However, we find this bound to be extremely loose in practice, and thus detrimental to performance.

Another approach to a lower bound is to note that sub-multiplicativity of the matrix norm gives

$$\left\|\mathbf{H}^2\mathbf{g}\right\| \leq \left\|\mathbf{H}^2\right\|\,\|\mathbf{g}\| \tag{45}$$

$$\implies \left\|\mathbf{H}^2\right\| \geq \frac{\left\|\mathbf{H}^2\mathbf{g}\right\|}{\|\mathbf{g}\|}. \tag{46}$$

Since our algorithm already computes $\mathbf{H}^2\mathbf{g}$, this allows us to efficiently compute a lower bound on $V$ based on our condition:

$$V \geq \frac{\left\|\mathbf{H}^2\mathbf{g}\right\|}{\|\mathbf{g}\|}. \tag{47}$$

In practice, the algorithm is initialised with some initial value of $V$ (specifically $V = 100$ in our experiments) which is then increased to $\frac{\left\|\mathbf{H}^2\mathbf{g}\right\|}{\|\mathbf{g}\|}$ whenever the bound in (47) is violated.

## D.2  Justification of the Truncated Series

We have shown that our infinite series (29) converges to the required transformed Hessian, but clearly we will be forced to truncate the series to $K$ terms in practical implementation. In this subsection, we informally justify the appropriateness of this truncation.

Restating (4),

$$(\mathbf{H}^2)^{-\frac{1}{2}}\mathbf{g} = \frac{1}{\sqrt{V}} \sum_{k=0}^{\infty} \frac{1}{2^{2k}} \binom{2k}{k} \left(\mathbf{I} - \frac{1}{V}\mathbf{H}^2\right)^k \mathbf{g}, \tag{48}$$

and recalling we denote the $k$th term of the summation by $\mathbf{a}_k$, we have from Algorithm 1 that

$$\mathbf{a}_0 = \mathbf{g}, \qquad \mathbf{a}_{k+1} = \frac{4k^2 - 2k}{4k^2}\left(\mathbf{I} - \frac{1}{V}\mathbf{H}^2\right)\mathbf{a}_k. \tag{49}$$

Now, for $k = 0, 1, 2, \cdots$, we have $\frac{4k^2 - 2k}{4k^2} < 1$, and we have $\left\|\mathbf{I} - \frac{1}{V}\mathbf{H}^2\right\| < 1$ by construction in order to secure convergence. It follows that

$$\left\|\left(\mathbf{I} - \frac{1}{V}\mathbf{H}^2\right)^k\right\| < 1 \tag{50}$$

for $k = 0, 1, 2, \cdots$. Thus, we have $\|\mathbf{a}_{k+1}\| < \|\mathbf{a}_k\|$ for such $k$, as suggested by the convergence property of our series, and we can describe the sequence of terms of the summation to be monotonically decreasing in magnitude. It is thus justifiable to suppose that, if we wish to take finitely many terms of the series, we should prioritise the earlier terms (smaller $k$), since these will have the greatest impact on the summation.

To develop further insight into this behaviour, recall we exploited the real, symmetric nature of $\mathbf{H}$ to eigendecompose it as $\mathbf{H} = \mathbf{Q}\mathbf{\Lambda}\mathbf{Q}^\mathsf{T}$. Substituting this decomposition into our series gives

$$(\mathbf{H}^2)^{-\frac{1}{2}}\mathbf{g} = \frac{1}{\sqrt{V}} \sum_{k=0}^{\infty} \frac{1}{2^{2k}} \binom{2k}{k} \left(\mathbf{I} - \frac{1}{V}\mathbf{H}^2\right)^k \mathbf{g} \tag{51}$$

$$= \frac{1}{\sqrt{V}} \sum_{k=0}^{\infty} \frac{1}{2^{2k}} \binom{2k}{k} \left(\mathbf{I} - \frac{1}{V}(\mathbf{Q}\mathbf{\Lambda}\mathbf{Q}^\mathsf{T})^2\right)^k \mathbf{g} \tag{52}$$

$$= \frac{1}{\sqrt{V}} \sum_{k=0}^{\infty} \frac{1}{2^{2k}} \binom{2k}{k} \left(\mathbf{Q}\mathbf{Q}^\mathsf{T} - \frac{1}{V}\mathbf{Q}\mathbf{\Lambda}^2\mathbf{Q}^\mathsf{T}\right)^k \mathbf{g} \tag{53}$$

$$= \frac{1}{\sqrt{V}} \sum_{k=0}^{\infty} \frac{1}{2^{2k}} \binom{2k}{k} \left(\mathbf{Q}\left(\mathbf{I} - \frac{1}{V}\mathbf{\Lambda}^2\right)\mathbf{Q}^\mathsf{T}\right)^k \mathbf{g} \tag{54}$$

$$= \frac{1}{\sqrt{V}}\mathbf{Q} \sum_{k=0}^{\infty} \frac{1}{2^{2k}} \binom{2k}{k} \left(\mathbf{I} - \frac{1}{V}\mathbf{\Lambda}^2\right)^k \mathbf{Q}^\mathsf{T}\mathbf{g}. \tag{55}$$

Since $\mathbf{I}$ and $\mathbf{\Lambda}$ are diagonal matrices, this series is actually a parallel combination of independent scalar series, and we can consider each diagonal component individually. For an arbitrary eigenvalue $\lambda$, this gives

$$\sum_{k=0}^{\infty} \frac{1}{2^{2k}} \binom{2k}{k} \left(1 - \frac{1}{V}\lambda^2\right)^k \tag{56}$$

Now, we specifically chose $V$ to be larger than the greatest eigenvalue magnitude of $\mathbf{H}^2$. Since we also assumed $\mathbf{H}^2$ has only positive eigenvalues, we can say $0 < 1 - \frac{1}{V}\lambda < 1$. This common ratio will be near zero for the largest eigenvalues $\lambda$, so we will see the most rapid convergence of these components of the series. Similarly, the common ratio will be near unity when $\lambda$ is near zero, so we will see the slowest convergence in these components.

This result allows us to consider the high- and low-eigenvalue components of the transformed Hessian independently. High-curvature directions in the space, indicated by large eigenvalues, will converge relatively quickly, so we expect the earlier terms of the series to be of most use in approximating these curvatures. As $k$ increases, the main contribution of each term is towards progressively smaller eigenvalues, representing lower-curvature regions of the space. Thus, the more-impactful higher-curvature information is addressed predominantly towards the start of the series, so even if we only consider finitely many terms, we can be sure none of the first $K$ terms could more optimally be replaced by a later term.

### D.3 Convergence and Escape

We follow the proof of Paternain et al. (2019) to prove that in the neighbourhood of a critical point, our method will converge to the critical point in locally convex directions and move away from the critical point in locally concave directions. Let $\mathbf{x}_C$ be any critical point, and define the immediate vicinity of $\mathbf{x}_C$ by the closed $\beta'$-ball $\mathcal{Q} = \{\mathbf{x} \in \mathbb{R}^P \mid \|\mathbf{x} - \mathbf{x}_C\| \leq \beta'\}$ for some $\beta'$. We require the following assumptions:

**Assumption 1.** *Over $\mathcal{Q}$, the loss function $f(\mathbf{x})$ is twice continuously differentiable, and further the gradient $\mathbf{g}(\mathbf{x})$ and Hessian $\mathbf{H}(\mathbf{x})$ are Lipschitz continuous. Specifically, there exist constants $M, L > 0$ such that for any $\mathbf{x}, \mathbf{y} \in \mathcal{Q} \subset \mathbb{R}^P$*

$$\|\mathbf{g}(\mathbf{x}) - \mathbf{g}(\mathbf{y})\| \leq M \|\mathbf{x} - \mathbf{y}\| \tag{57}$$

$$\|\mathbf{H}(\mathbf{x}) - \mathbf{H}(\mathbf{y})\| \leq L \|\mathbf{x} - \mathbf{y}\| \tag{58}$$

**Assumption 2.** *The hessian $\mathbf{H}(\mathbf{x})$ is invertible over $\mathcal{Q}$. Specifically, there exists a $\delta > 0$ such that $|\lambda_i(\mathbf{H}(\mathbf{x}))| > \delta$ for all $\mathbf{x} \in \mathcal{Q} \subset \mathbb{R}^P$ and $i = 1, 2, ..., P$. This additionally implies non-degeneracy of the saddle point.*

We note that Paternain et al. (2019) also require Assumption 1, though they assume a weaker form of Assumption 2, namely that the $|\lambda_i(\mathbf{H}(\mathbf{x}))| > \delta$ must hold at all local minima and saddle points, rather than in a $\beta'$-ball around local minima and saddle points. In practice, applying damping to the Hessian ensures that this assumption holds.

We also assume that our series approximation to the inverted saddle-free Hessian in Equation (4) has converged. We use the notation $\overline{\mathbf{A}}$ to denote the matrix obtained by taking the absolute value of each eigenvalue of $\mathbf{A}$ and note that the saddle-free Hessian, $\overline{\mathbf{H}}$ is thus written as $\overline{\mathbf{H}}(\mathbf{x})^{-1} = \left(\mathbf{Q}\overline{\mathbf{\Lambda}}\mathbf{Q}^{\mathsf{T}}\right)^{-1}$, where $\mathbf{Q}\mathbf{\Lambda}\mathbf{Q}^{\mathsf{T}}$ is the eigendecomposition of $\mathbf{H}(\mathbf{x})$. Without loss of generality, we shall assume the eigenvalues to be arranged in ascending order, such that $\lambda_1 \leq \lambda_2 \leq \cdots \leq \lambda_P$ and the $i$th column of $\mathbf{Q}$ is the eigenvector associated with eigenvalue $\lambda_i$.

Recall the critical point of interest is $\mathbf{x}_C$. We let $\mathbf{g}_+(\mathbf{x})$ denote the gradient at $\mathbf{x}$ projected onto the subspace of $\mathbf{H}(\mathbf{x}_C)$'s eigenvectors associated with the positive eigenvalues of $\mathbf{H}(\mathbf{x}_C)$. Similarly, let $\mathbf{g}_-(\mathbf{x})$ denote the projection of $\mathbf{g}(\mathbf{x})$ onto the subspace defined by the eigenvectors of $\mathbf{H}(\mathbf{x}_C)$ associated with negative eigenvalues. We now go on to prove that given the assumptions above, for a point $\mathbf{x}_t$ that is in the neighbourhood $\mathcal{Q}$ of a critical point $\mathbf{x}_C$, our method will converge in the subspace corresponding to the positive eigenvalues of $\mathbf{H}(\mathbf{x}_C)$ and escape in the subspace corresponding to the negative eigenvalues of $\mathbf{H}(\mathbf{x}_C)$. In other words, we show that $\|\mathbf{g}_+(\mathbf{x}_{t+1})\|$ converges to zero and that $\|\mathbf{g}_-(\mathbf{x}_{t+1})\|$ will grow.

**Theorem 1.** *Given Assumptions 1 and 2, suppose that $\|\mathbf{x}_t - \mathbf{x}_C\| < \beta \|\mathbf{g}(\mathbf{x}_t)\|$ where $\mathbf{x}_C$ is a critical point and let $\overline{\mathbf{H}}(\mathbf{x})$, $\mathbf{g}_+(\mathbf{x})$ and $\mathbf{g}_-(\mathbf{x})$ be defined as above. Let $\|\cdot\|$ denote a sub-multiplicative norm. Then both the following inequalities hold:*

$$\|\mathbf{g}_+(\mathbf{x}_{t+1})\| \leq D \|\mathbf{g}(\mathbf{x}_t)\|^2 \tag{59}$$

$$\|\mathbf{g}_-(\mathbf{x}_{t+1})\| \geq 2 \|\mathbf{g}_-(\mathbf{x}_t)\| - D \|\mathbf{g}(\mathbf{x}_t)\|^2 \tag{60}$$

*where $D = \frac{LC_\delta^2 + 4LC_\delta\beta}{2}$.*

*Proof.* We split the the proof into two cases, one for positive eigenvalues, corresponding to (59) and one for negative eigenvalues, corresponding to (60). We start with the negative case.

*Case 1: Negative Eigenvalues*

Noting that $\mathbf{g}(\mathbf{x}_t + \theta\Delta\mathbf{x})$ is an anti-derivative of $\mathbf{H}(\mathbf{x}_t + \theta\Delta\mathbf{x})\Delta\mathbf{x}$ with respect to $\theta$, we can write

$$\mathbf{g}(\mathbf{x}_{t+1}) = \mathbf{g}(\mathbf{x}_t) + \int_0^1 \mathbf{H}(\mathbf{x}_t + \theta\Delta\mathbf{x})\Delta\mathbf{x}\, d\theta, \tag{61}$$

where $\Delta\mathbf{x} = \mathbf{x}_{t+1} - \mathbf{x}_t$. Now, the update rule of our method is given by

$$\mathbf{x}_{t+1} = \mathbf{x}_t - \overline{\mathbf{H}}(\mathbf{x}_t)^{-1}\mathbf{g}(\mathbf{x}_t), \tag{62}$$

so that $-\overline{\mathbf{H}}(\mathbf{x}_t)^{-1}\mathbf{g}(\mathbf{x}_t) = \Delta\mathbf{x}$. Using this fact, we note that $\mathbf{g}(\mathbf{x}_t) = \overline{\mathbf{H}}(\mathbf{x}_t)\overline{\mathbf{H}}(\mathbf{x}_t)^{-1}\mathbf{g}(\mathbf{x}_t) = -\overline{\mathbf{H}}(\mathbf{x}_t)\Delta\mathbf{x}$. We add and subtract $\mathbf{g}(\mathbf{x}_t)$ from (61) as follows:

$$
\begin{aligned}
\mathbf{g}(\mathbf{x}_{t+1}) &= \mathbf{g}(\mathbf{x}_t) + \mathbf{g}(\mathbf{x}_t) - \mathbf{g}(\mathbf{x}_t) + \int_0^1 \mathbf{H}(\mathbf{x}_t + \theta\Delta\mathbf{x})\Delta\mathbf{x}\, d\theta \\
&= 2\mathbf{g}(\mathbf{x}_t) + \overline{\mathbf{H}}(\mathbf{x}_t)\Delta\mathbf{x} + \int_0^1 \mathbf{H}(\mathbf{x}_t + \theta\Delta\mathbf{x})\Delta\mathbf{x}\, d\theta \\
&= 2\mathbf{g}(\mathbf{x}_t) + \int_0^1 \left(\mathbf{H}(\mathbf{x}_t + \theta\Delta\mathbf{x}) + \overline{\mathbf{H}}(\mathbf{x}_t)\right)\Delta\mathbf{x}\, d\theta.
\end{aligned}
\tag{63}
$$

We continue in the manner of Paternain et al. (2019) to add and subtract $\mathbf{H}(\mathbf{x}_C)$, $\mathbf{H}(\mathbf{x}_t)$, and $\overline{\mathbf{H}}(\mathbf{x}_C)$ inside the integral and shuffle the terms to arrive at:

$$\mathbf{g}(\mathbf{x}_{t+1}) = 2\mathbf{g}(\mathbf{x}_t) + \int_0^1 \left(\mathbf{H}(\mathbf{x}_t + \theta\Delta\mathbf{x}) - \mathbf{H}(\mathbf{x}_t)\right) \Delta\mathbf{x}\, d\theta$$

$$+ \int_0^1 \left(\mathbf{H}(\mathbf{x}_t) - \mathbf{H}(\mathbf{x}_C)\right) \Delta\mathbf{x}\, d\theta + \int_0^1 \left(\overline{\mathbf{H}}(\mathbf{x}_t) - \overline{\mathbf{H}}(\mathbf{x}_C)\right) \Delta\mathbf{x}\, d\theta$$

$$+ \int_0^1 \left(\mathbf{H}(\mathbf{x}_C) + \overline{\mathbf{H}}(\mathbf{x}_C)\right) \Delta\mathbf{x}\, d\theta$$

$$= 2\mathbf{g}(\mathbf{x}_t) + \int_0^1 \left(\mathbf{H}(\mathbf{x}_t + \theta\Delta\mathbf{x}) - \mathbf{H}(\mathbf{x}_t)\right) \Delta\mathbf{x}\, d\theta$$

$$+ \left(\mathbf{H}(\mathbf{x}_t) - \mathbf{H}(\mathbf{x}_C)\right)\Delta\mathbf{x} + \left(\overline{\mathbf{H}}(\mathbf{x}_t) - \overline{\mathbf{H}}(\mathbf{x}_C)\right)\Delta\mathbf{x} + \left(\mathbf{H}(\mathbf{x}_C) + \overline{\mathbf{H}}(\mathbf{x}_C)\right)\Delta\mathbf{x}. \tag{64}$$

Let $\mathbf{Q}_-$ denote the matrix of eigenvectors corresponding to negative eigenvalues of $\mathbf{H}(\mathbf{x}_C)$. We pre-multiply the left and right of Equation (64) by $\mathbf{Q}_-^\mathsf{T}$ and consider each of the last four terms separately.

For the integrand, we note that $\left\|\mathbf{Q}_-^\mathsf{T}\right\| \leq 1$ since the columns are normalised eigenvectors. Moreover, since $\mathbf{H}(\mathbf{x})$ is Lipschitz by Assumption 1, we have that $\|\mathbf{H}(\mathbf{x}_t + \theta\Delta\mathbf{x}) - \mathbf{H}(\mathbf{x}_t)\| \leq L \|\mathbf{x}_t + \theta\Delta\mathbf{x} - \mathbf{x}_t\| = L\theta \|\Delta\mathbf{x}\|$ so that

$$\left\|\mathbf{Q}_-^\mathsf{T}\left(\mathbf{H}(\mathbf{x}_t + \theta\Delta\mathbf{x}) - \mathbf{H}(\mathbf{x}_t)\right)\Delta\mathbf{x}\right\| \leq \|\mathbf{H}(\mathbf{x}_t + \theta\Delta\mathbf{x}) - \mathbf{H}(\mathbf{x}_t)\| \|\Delta\mathbf{x}\| \leq \theta L \|\Delta\mathbf{x}\|^2. \tag{65}$$

We handle the next two terms in a similar way, applying the Lipschitz assumption:

$$\left\|\mathbf{Q}_-^\mathsf{T}\left(\mathbf{H}(\mathbf{x}_t) - \mathbf{H}(\mathbf{x}_C)\right)\Delta\mathbf{x}\right\| \leq \|\mathbf{H}(\mathbf{x}_t) - \mathbf{H}(\mathbf{x}_C)\| \|\Delta\mathbf{x}\| \leq L \|\mathbf{x}_t - \mathbf{x}_C\| \|\Delta\mathbf{x}\|, \tag{66}$$

$$\left\|\mathbf{Q}_-^\mathsf{T}\left(\overline{\mathbf{H}}(\mathbf{x}_t) - \overline{\mathbf{H}}(\mathbf{x}_C)\right)\Delta\mathbf{x}\right\| \leq \left\|\overline{\mathbf{H}}(\mathbf{x}_t) - \overline{\mathbf{H}}(\mathbf{x}_C)\right\| \|\Delta\mathbf{x}\| \leq L \|\mathbf{x}_t - \mathbf{x}_C\| \|\Delta\mathbf{x}\|. \tag{67}$$

Finally, we show that the last term in (64) becomes zero. Using the eigendecomposition of $\mathbf{H}(\mathbf{x}_C)$, we observe that

$$\mathbf{Q}_-^\mathsf{T}\left(\mathbf{H}(\mathbf{x}_C) + \overline{\mathbf{H}}(\mathbf{x}_C)\right) = \mathbf{Q}_-^\mathsf{T}\left(\mathbf{Q}\mathbf{\Lambda}(\mathbf{x}_C)\mathbf{Q}^\mathsf{T} + \mathbf{Q}\overline{\mathbf{\Lambda}}(\mathbf{x}_C)\mathbf{Q}^\mathsf{T}\right) = \mathbf{Q}_-^\mathsf{T}\mathbf{Q}\left(\mathbf{\Lambda}(\mathbf{x}_C) + \overline{\mathbf{\Lambda}}(\mathbf{x}_C)\right)\mathbf{Q}^\mathsf{T}$$

Suppose there are $d$ negative eigenvalues. Then $\mathbf{Q}_-^\mathsf{T}\mathbf{Q} = [\mathbf{I}_d, \mathbf{0}_{d \times m-d}]$, i.e. the eigenvectors of $\mathbf{Q}$ corresponding to positive eigenvalues are mapped to zero, and those corresponding to negative eigenvalues are mapped to a unit basis vector. This is because the columns of $\mathbf{Q}$ are orthonormal, so the inner product of columns is unity if the columns are equal and zero otherwise. Furthermore, $\mathbf{\Lambda}(\mathbf{x}_C) + \overline{\mathbf{\Lambda}}(\mathbf{x}_C)$ is diagonal where the first $d$ elements are zero and the remaining elements double (due to negative eigenvalues cancelling out with their positive counterparts in $\overline{\mathbf{\Lambda}}(\mathbf{x}_C)$ and positive eigenvalues being added to their positive counterparts in $\overline{\mathbf{\Lambda}}(\mathbf{x}_C)$). But then the product $\mathbf{Q}_-^\mathsf{T}\mathbf{Q}\left(\mathbf{\Lambda}(\mathbf{x}_C) + \overline{\mathbf{\Lambda}}(\mathbf{x}_C)\right) = \mathbf{0}$, because the zero components of each term complement each other.

We recall the following identity from the reverse triangle inequality: $\|a + b\| = \|a - (-b)\| \geq |\|a\| - \|b\|| \geq \|a\| - \|b\|$ and combine it with (65), (66) and (67) to lower bound Equation (64) as follows:

$$\|\mathbf{g}_-(\mathbf{x}_{t+1})\| \geq 2 \|\mathbf{g}_-(\mathbf{x}_t)\| - L \|\Delta\mathbf{x}\|^2 \int_0^1 \theta\, d\theta - 2L \|\mathbf{x}_t - \mathbf{x}_C\| \|\Delta\mathbf{x}\|. \tag{68}$$

We use the definition of the update step to bound $\|\Delta\mathbf{x}\|$ as follows:

$$\mathbf{x}_{t+1} - \mathbf{x}_t = -\overline{\mathbf{H}}(\mathbf{x}_t)^{-1}\mathbf{g}(\mathbf{x}_t) \text{ so that}$$

$$\|\Delta\mathbf{x}\| \leq \left\|\overline{\mathbf{H}}(\mathbf{x}_t)^{-1}\right\| \|\mathbf{g}(\mathbf{x}_t)\|$$

$$= \left\|(\mathbf{Q}\overline{\mathbf{\Lambda}}\mathbf{Q}^\mathsf{T})^{-1}\right\| \|\mathbf{g}(\mathbf{x}_t)\|$$

$$\leq \left\|\overline{\mathbf{\Lambda}}^{-1}\right\| \|\mathbf{g}(\mathbf{x}_t)\| \tag{69}$$

Now, $\left\|\overline{\mathbf{\Lambda}}^{-1}\right\|$ is bounded because $\max_{i=1..P} \lambda(\overline{\mathbf{\Lambda}}^{-1}) \leq \frac{1}{\delta}$ by Assumption 2[4]. For $\|\cdot\|_2$, this bound is $\left\|\overline{\mathbf{\Lambda}}^{-1}\right\| \leq \frac{1}{\delta}$. In the general case, we denote the bound by $C_\delta$ (where $C_\delta$ may also depend on the dimensionality of the problem for some choices of $\|\cdot\|$). This, along with our assumption that $\mathbf{x}_t$ is near $\mathbf{x}_C$ gives us the final bound:

$$
\begin{aligned}
\|\mathbf{g}_-(\mathbf{x}_{t+1})\| &\geq 2\|\mathbf{g}_-(\mathbf{x}_t)\| - \frac{LC_\delta^2}{2}\|\mathbf{g}(\mathbf{x}_t)\|^2 - 2LC_\delta\beta\|\mathbf{g}(\mathbf{x}_t)\|^2 \\
&\geq 2\|\mathbf{g}_-(\mathbf{x}_t)\| - \left(\frac{LC_\delta^2 + 4LC_\delta\beta}{2}\right)\|\mathbf{g}(\mathbf{x}_t)\|^2.
\end{aligned}
\tag{70}
$$

*Case 2: Positive Eigenvalues*

As in the negative case, we start with

$$
\mathbf{g}(\mathbf{x}_{t+1}) = \mathbf{g}(\mathbf{x}_t) + \int_0^1 \mathbf{H}(\mathbf{x}_t + \theta\Delta\mathbf{x})\Delta\mathbf{x}\,d\theta.
\tag{71}
$$

This time, we substitute $\mathbf{g}(\mathbf{x}_t) = \overline{\mathbf{H}}(\mathbf{x}_t)\overline{\mathbf{H}}(\mathbf{x}_t)^{-1}\mathbf{g}(\mathbf{x}_t) = -\overline{\mathbf{H}}(\mathbf{x}_t)\Delta\mathbf{x}$ directly to obtain

$$
\mathbf{g}(\mathbf{x}_{t+1}) = \int_0^1 \left(\mathbf{H}(\mathbf{x}_t + \theta\Delta\mathbf{x}) - \overline{\mathbf{H}}(\mathbf{x}_t)\right)\Delta\mathbf{x}\,d\theta.
\tag{72}
$$

We proceed as in the negative case, adding and subtracting $\mathbf{H}(\mathbf{x}_C)$, $\mathbf{H}(\mathbf{x}_t)$, and $\overline{\mathbf{H}}(\mathbf{x}_C)$ to obtain

$$
\begin{aligned}
\mathbf{g}(\mathbf{x}_{t+1}) = \int_0^1 \left(\mathbf{H}(\mathbf{x}_t + \theta\Delta\mathbf{x}) - \mathbf{H}(\mathbf{x}_t)\right)\Delta\mathbf{x}\,d\theta + \left(\mathbf{H}(\mathbf{x}_t) - \mathbf{H}(\mathbf{x}_C)\right)\Delta\mathbf{x} \\
+ \left(\overline{\mathbf{H}}(\mathbf{x}_C) - \overline{\mathbf{H}}(\mathbf{x}_t)\right)\Delta\mathbf{x} + \left(\mathbf{H}(\mathbf{x}_C) - \overline{\mathbf{H}}(\mathbf{x}_C)\Delta\mathbf{x},
\end{aligned}
\tag{73}
$$

noting that the last two terms are different to the negative case. Let $\mathbf{Q}_+$ denote the matrix of eigenvectors corresponding to positive eigenvalues of $\mathbf{H}(\mathbf{x}_C)$. Multiply the left and the right hand side of Equation (73) by $\mathbf{Q}_+^{\mathsf{T}}$ and apply the triangle equality to obtain

$$
\begin{aligned}
\|\mathbf{g}_+(\mathbf{x}_{t+1})\| &\leq \left\|\int_0^1 \left(\mathbf{H}(\mathbf{x}_t + \theta\Delta\mathbf{x}) - \mathbf{H}(\mathbf{x}_t)\right)\Delta\mathbf{x}\,d\theta\right\| + \left\|\left(\mathbf{H}(\mathbf{x}_t) - \mathbf{H}(\mathbf{x}_C)\right)\Delta\mathbf{x}\right\| \\
&\quad + \left\|\left(\overline{\mathbf{H}}(\mathbf{x}_C) - \overline{\mathbf{H}}(\mathbf{x}_t)\right)\Delta\mathbf{x}\right\| + \left\|\left(\mathbf{H}(\mathbf{x}_C) - \overline{\mathbf{H}}(\mathbf{x}_C)\Delta\mathbf{x}\right\|.
\end{aligned}
\tag{74}
$$

Using Equations (65), (66) and (67) as in the negative case, we can bound the first three terms. The bound on the last term follows similar reasoning as before:

$$
\mathbf{Q}_+^{\mathsf{T}}\left(\mathbf{H}(\mathbf{x}_C) - \overline{\mathbf{H}}(\mathbf{x}_C)\right) = \mathbf{Q}_+^{\mathsf{T}}\mathbf{Q}\left(\mathbf{\Lambda}(\mathbf{x}_C) - \overline{\mathbf{\Lambda}}(\mathbf{x}_C)\right)\mathbf{Q}^{\mathsf{T}}
\tag{75}
$$

where $\mathbf{Q}_+^{\mathsf{T}}\mathbf{Q}\left(\mathbf{\Lambda}(\mathbf{x}_C) - \overline{\mathbf{\Lambda}}(\mathbf{x}_C)\right) = \mathbf{0}$ because $\mathbf{Q}_+^{\mathsf{T}}\mathbf{Q}$ maps the eigenvectors of $\mathbf{Q}$ that correspond to negative eigenvalues to zero and $\mathbf{\Lambda}(\mathbf{x}_C) - \overline{\mathbf{\Lambda}}(\mathbf{x}_C)$ produces a diagonal matrix where the positive eigenvalues cancel out and the negative eigenvalues double.

We thus arrive at the bound

$$
\begin{aligned}
\|\mathbf{g}_+(\mathbf{x}_{t+1})\| &\leq \frac{LC_\delta^2}{2}\|\mathbf{g}(\mathbf{x}_t)\|^2 + 2LC_\delta\beta\|\mathbf{g}(\mathbf{x}_t)\|^2 \\
&\leq \left(\frac{LC_\delta^2 + 4LC_\delta\beta}{2}\right)\|\mathbf{g}(\mathbf{x}_t)\|^2.
\end{aligned}
\tag{76}
$$

$\square$

---

[4]While we do not consider it in this work, we note that the use of canonical second-order damping methods, which replace a curvature matrix $\mathbf{A}$ by $\mathbf{A} + \lambda\mathbf{I}$ and thus increase every eigenvalue of $\mathbf{A}$ by $\lambda$, allows us to relax Assumption 2 to hold for the *damped* (saddle-free) Hessian, and thus admit arbitrary Hessians by suitable choice of $\lambda$.

Reprising the arguments in Corollary 3.3 and Proposition 3.4 of Paternain et al. (2019), (59) gives that $\|\mathbf{g}_+(\mathbf{x}_{t+1})\|$ converges quadratically to zero if the greatest contribution to $\|\mathbf{g}(\mathbf{x}_t)\| = \|\mathbf{g}_+(\mathbf{x}_t) + \mathbf{g}_-(\mathbf{x}_t)\|$ is from the $\mathbf{g}_+(\mathbf{x}_t)$ term (as for a local minimum), and (60) gives that $\|\mathbf{g}_-(\mathbf{x}_t)\|$ grows by a multiplicative factor $2 - D$ (where we may choose the free parameter $D$ such that $0 < D < 1$) if $\|\mathbf{g}(\mathbf{x}_t)\|^2$ is negligible compared to $\|\mathbf{g}_-(\mathbf{x}_t)\|$. In combination, these results justify our claim to converge to local minima and repel saddle points.

### D.4 Rate of Convergence

In this subsection, we provide a brief analysis of the rate of convergence of our algorithm to critical points, following a similar proof pattern to that of classical Newton methods. Throughout, we will assume every term of our modified-Hessian summation is used, such that our Hessian transformation is exact.

For brevity, denote by $\overline{\mathbf{H}}$ the matrix obtained by taking the absolute value of every eigenvalue of $\mathbf{H}$. With this shorthand, recall the exact version of our update rule is

$$\mathbf{x}_{t+1} = \mathbf{u}(\mathbf{x}_t) = \mathbf{x}_t - \overline{\mathbf{H}}(\mathbf{x}_t)^{-1}\mathbf{g}(\mathbf{x}_t), \tag{77}$$

where we will now explicitly denote the points at which the Hessian $\mathbf{H}$ and gradient $\mathbf{g}$ are calculated.

Let $\mathbf{x}_C$ be an arbitrary critical point of the objective function $f$, such that $\mathbf{g}(\mathbf{x}_C) = \mathbf{0}$. The latter fact gives $\mathbf{u}(\mathbf{x}_C) = \mathbf{x}_C$, and thus $\mathbf{x}_t - \mathbf{x}_C = \mathbf{u}(\mathbf{x}_{t-1}) - \mathbf{u}(\mathbf{x}_C)$.

Consider taking a Taylor expansion of our update rule $\mathbf{u}(\mathbf{x})$ about $\mathbf{x}_C$:

$$\mathbf{u}(\mathbf{x}_t) = \mathbf{u}(\mathbf{x}_C) + \nabla\mathbf{u}(\mathbf{x}_C)^\mathsf{T}\underbrace{(\mathbf{x}_t - \mathbf{x}_C)}_{\boldsymbol{\epsilon}_t} + \frac{1}{2}(\mathbf{x}_t - \mathbf{x}_C)^\mathsf{T}\nabla\nabla\mathbf{u}(\mathbf{x}_C)(\mathbf{x}_t - \mathbf{x}_C) + \cdots. \tag{78}$$

Denote by $\boldsymbol{\epsilon}_t = \mathbf{x}_t - \mathbf{x}_C$ the error between our critical point and $\mathbf{x}_t$. Note this is unrelated to any discussion of Wynn's $\epsilon$-algorithm (Wynn, 1956a); we have chosen to reflect standard notation by overloading $\boldsymbol{\epsilon}$ here.

Now, by direct differentiation of (77), we have

$$\nabla\mathbf{u}(\mathbf{x}_C) = \mathbf{I} - \nabla\overline{\mathbf{H}}(\mathbf{x}_C)^{-1}\underbrace{\mathbf{g}(\mathbf{x}_C)}_{\mathbf{0}} - \overline{\mathbf{H}}(\mathbf{x}_C)^{-1}\mathbf{H}(\mathbf{x}_C). \tag{79}$$

Noting that $\mathbf{H}$ and $\overline{\mathbf{H}}$ have the same eigenvectors, we may gain insight into the final product by eigendecomposing it:

$$\overline{\mathbf{H}}(\mathbf{x}_C)^{-1}\mathbf{H}(\mathbf{x}_C) = \mathbf{Q}\overline{\boldsymbol{\Lambda}}^{-1}\mathbf{Q}^\mathsf{T}\mathbf{Q}\boldsymbol{\Lambda}\mathbf{Q}^\mathsf{T} \tag{80}$$

$$= \mathbf{Q}\overline{\boldsymbol{\Lambda}}^{-1}\boldsymbol{\Lambda}\mathbf{Q}^\mathsf{T}. \tag{81}$$

Since $|\lambda|$ and $\lambda$ are corresponding eigenvalues of $\overline{\mathbf{H}}$ and $\mathbf{H}$, the result of this product is a matrix with the same eigenvectors as $\mathbf{H}$, but with eigenvalues $\frac{\lambda}{|\lambda|} = \operatorname{sign}\lambda$. Consequently, we recover different dynamics for positive and negative eigenvalues — equivalently, positive and negative curvatures — in the space (recall our assumption of invertibility of $\mathbf{H}$ provides $\lambda \neq 0$).

We proceed to analyse each case individually, effectively creating two complementary subspaces of the optimisation space. We will use the subscripts $+$ and $-$ to denote the positive- and negative-curvature subspaces, respectively. Note that the orthogonality of these subspaces (ensured by the real, symmetric nature of $\mathbf{H}$ giving orthogonal eigenvectors) justifies our independent analysis.

For the positive-curvature subspace, $\overline{\boldsymbol{\Lambda}}_+ = \boldsymbol{\Lambda}_+$, whence $\overline{\mathbf{H}}_+(\mathbf{x}_C)^{-1}\mathbf{H}_+(\mathbf{x}_C) = \mathbf{I}$. This gives

$$\nabla\mathbf{u}_+(\mathbf{x}_C) = \mathbf{I} - \mathbf{I} = \mathbf{0}, \tag{82}$$

which collapses our Taylor series to

$$\mathbf{u}_+(\mathbf{x}_t) = \mathbf{u}_+(\mathbf{x}_C) + \frac{1}{2}\boldsymbol{\epsilon}_{t,+}^\mathsf{T}\nabla\nabla\mathbf{u}_+(\mathbf{x}_C)\boldsymbol{\epsilon}_{t,+} + \cdots \tag{83}$$

$$\implies \mathbf{u}_+(\mathbf{x}_t) - \mathbf{u}_+(\mathbf{x}_C) = \boldsymbol{\epsilon}_{t+1,+} = \mathcal{O}(\boldsymbol{\epsilon}_{t,+}^2), \tag{84}$$

where by $\mathcal{O}(\boldsymbol{\epsilon}_{t,+}^2)$ we mean to indicate that the positive-subspace error between our current point and a critical point varies quadratically with time, as the truncated terms are in higher-order products of $\boldsymbol{\epsilon}_{t,+}$.

We go on to repeat this argument for the negative-curvature subspace where $\overline{\boldsymbol{\Lambda}}_- = -\boldsymbol{\Lambda}_-$, so that $\overline{\mathbf{H}}_-(\mathbf{x}_C)^{-1}\mathbf{H}_-(\mathbf{x}_C) = -\mathbf{I}$. Then, we recover

$$\nabla\mathbf{u}_-(\mathbf{x}_C) = \mathbf{I} + \mathbf{I} = 2\mathbf{I}, \tag{85}$$

which collapses our Taylor series in a different way:

$$\mathbf{u}_-(\mathbf{x}_t) = \mathbf{u}_-(\mathbf{x}_C) + 2\boldsymbol{\epsilon}_{t,-} + \frac{1}{2}\boldsymbol{\epsilon}_{t,-}^{\mathsf{T}}\nabla\nabla\mathbf{u}_-(\mathbf{x}_C)\boldsymbol{\epsilon}_{t,-} + \cdots \tag{86}$$

$$\implies \mathbf{u}_-(\mathbf{x}_t) - \mathbf{u}_-(\mathbf{x}_C) = \boldsymbol{\epsilon}_{t+1,-} = 2\boldsymbol{\epsilon}_t + \mathcal{O}(\boldsymbol{\epsilon}_{t,-}^2); \tag{87}$$

that is, that the negative-subspace error between our current point and a critical point diverges exponentially with time.

This derivation proves that, over time, our algorithm will converge to some critical point $\mathbf{x}_C$. But our derivation in Appendix D.3 shows that our algorithm *escapes* from non-degenerate saddle points and local maxima, and is *attracted* to local minima. Thus, any convergence to a critical point *must* be to a local minimum; the results of these two sections combine to give that our algorithm converges quadratically along positive-curvature directions and escapes exponentially from negative-curvature directions.

