# OpenReview forum: "Series of Hessian-Vector Products for Tractable Saddle-Free Newton Optimisation of Neural Networks"
_TMLR — Accepted by TMLR_

### Review · Reviewer_2zn1 · 2023-11-24

**Summary Of Contributions:**

This paper presents a novel second-order approach to escaping saddle points. Unlike Saddle-Free Newton (SFN) methods, this approach computes updating directions by leveraging the inverse of the squared and principal square-rooted Hessian through an infinite series. Notably, it achieves this without necessitating eigenvalue decomposition of the Hessian matrix. The authors then introduce a truncated Hessian-free algorithm, utilizing series acceleration to approximate these directions. Experimental results demonstrate that the proposed algorithm performs comparably to SGD, Adam, KFAC and the exact SFN.

**Audience:**

Yes

**Claims And Evidence:**

Yes

**Requested Changes:**

LiSSA [4] also utilizes an infinite series to compute second-order updating directions, subsequently approximating them via truncation. Some discussion on this approach would be beneficial.

The convergence rate of Lanczos methods is well-documented [5] and can be enhanced by applying reorthogonalization for numerical stability. Additionally, efficient Hessian-free Newton methods like Newton-CG [2] and Newton-MR [3] variants employ negative (non-positive) curvature detections with complexity guarantees, enabling efficient escape from saddle points. It would be valuable to include discussion on these methods.

[1] https://proceedings.mlr.press/v70/carmon17a/carmon17a.pdf

[2] https://link.springer.com/article/10.1007/s10107-019-01362-7

[3] https://arxiv.org/abs/2208.07095

[4] https://www.jmlr.org/papers/volume18/16-491/16-491.pdf

[5] https://www-users.cse.umn.edu/~saad/eig_book_2ndEd.pdf

**Strengths And Weaknesses:**

Strengths：

The paper maintains a smooth flow, complemented by clear illustrations, rendering it easy to read.

The methods employed to approximate the infinite series for obtaining reasonable updating directions are commendable.


Weaknesses：

While the paper concentrates on proposing saddle-free algorithms, there is limited discussion on other saddle-free algorithms that utilize curvature information without necessitating eigenvalue decomposition, e.g., [1], [2], [3]. Especially, the experiments lack thorough comparisons in this regard.

The performance of the proposed algorithm does not consistently outperform the compared algorithms.

---

> ### Author Response · Authors · 2024-01-23
> **Response to Reviewer 2zn1**
>
> Thank you very much for your feedback on our work! We'd like to address some of your observations below:
>
> ### Missing coverage of LiSSA and other SFN algorithms
> Thank you for bringing our attention to the LiSSA, AGD-until-Guilty, Newton-CG and Newton-MR methods – we will include them in our literature review.
>
> ### Proposed algorithm does not consistently outperform benchmarks
> While our proposed algorithm does not consistently outperform benchmarks, we frame this work as a study of the power series approach to SFN methods and the resulting performance we can expect, which is nonetheless comparable to the benchmarks in many settings. Since we do not claim to outperform the benchmarks, we feel this is an informative contribution as our algorithm, even if not always an improvement on the baselines, is not much worse.
>
> ### Missing discussion of convergence and enhancement of Lanczos methods
> Thank you for pointing out this oversight. We will change our reference to the Lanczos algorithm to clarify that careful implementations can succeed.

---

### Review · Reviewer_DFUg · 2023-12-21

**Summary Of Contributions:**

In the paper, the new second-order optimization method is presented for non-convex optimization with applications to neural networks. The main idea of the method is using that $|A|=\sqrt{A^2}$ and applying the series technique to compute its inverse. The extensive experimental results are presented for big variety of optimization setups.

**Audience:**

Yes

**Claims And Evidence:**

No

**Requested Changes:**

* I would recommend adding the clear contribution subsection with the main claims from the paper.
* I would recommend adding Cubic Regularized Newton and Trust Region Newton to the literature review.
* I would recommend adding some versions of Trust-Region-CG and Cubic Newton + GD to the practical comparison.
* In Figure 3, KFAC(Kazuki) seems to ruin the visualization of the results, it is clear that it somehow diverged, but it took almost the whole space in Figure 3.

**Strengths And Weaknesses:**

Let me begin by highlighting the strengths:
* I like the idea of $|A|=\sqrt{A^2}$; it is well-motivated as it helps to not be attracted to the first-order stationary points.
* The experiments are performed and presented in the professional way with multiple seed fine-tuning and providing detailed specifications, which should help to the reproducibility of the results.

However, I would like to point out some weaknesses and limitations from my perspective:
* The main problem for me is the absence of a comparison with the main second-order methods for non-convex optimization: cubic regularized Newton [1-5] and Trust-region Newton [7-8]. Both of these methods have a long history of research, they are both probably optimal for non-convex functions. Both of them have efficient first-order subsolvers: Trust-Region-CG [9] and Cubic Newton + GD [6]. Both of them could escape from the first-order stationary point.  From this point, some claims in the paper are too strong and misleading, for example: “We propose an optimisation algorithm which addresses both of these concerns — to our knowledge, the first efficiently-scalable optimisation algorithm to asymptotically use the exact (eigenvalue-modified) inverse Hessian”.
* There is no convergence theorems, which would show the convergence rate of the method.
* From the practical point of view, it seems that the method has too many parameters and additional heuristics to perform comparably with the much more simple methods. “Our implementation of Algorithm 1, using tuned learning rate, momentum, series length K and order of acceleration N”. 4 parameters in total. In the main figures 2-3, the fine-tuned 4 parameters method with momentum and Hessian-vector information is losing to SGD with 1 parameter. This questions the practical performance of the proposed method for me.

[1] - Nesterov, Yurii, and Boris T. Polyak. "Cubic regularization of Newton method and its global performance." Mathematical Programming 108.1 (2006): 177-205.

[2] - Tripuraneni, Nilesh, et al. "Stochastic cubic regularization for fast nonconvex optimization." Advances in neural information processing systems 31 (2018).

[3] - Cartis, Coralia, Nicholas IM Gould, and Philippe L. Toint. "Adaptive cubic regularisation methods for unconstrained optimization. Part I: motivation, convergence and numerical results." Mathematical Programming 127.2 (2011): 245-295.

[4] - Wang, Zhe, et al. "Cubic regularization with momentum for nonconvex optimization." Uncertainty in Artificial Intelligence. PMLR, 2020.

[5] - Wang, Zhe, et al. "Stochastic variance-reduced cubic regularization for nonconvex optimization." The 22nd International Conference on Artificial Intelligence and Statistics. PMLR, 2019.

[6] - Carmon, Yair, and John Duchi. "Gradient descent finds the cubic-regularized nonconvex Newton step." SIAM Journal on Optimization 29.3 (2019): 2146-2178.

[7] - Conn, Andrew R., Nicholas IM Gould, and Philippe L. Toint. Trust region methods. Society for Industrial and Applied Mathematics, 2000.

[8] - Nocedal, Jorge, and Stephen J. Wright, eds. Numerical optimization. New York, NY: Springer New York, 1999.

[9] - Curtis, Frank E., et al. "Trust-region Newton-CG with strong second-order complexity guarantees for nonconvex optimization." SIAM Journal on Optimization 31.1 (2021): 518-544.

---

> ### Author Response · Authors · 2024-01-23
> **Response to Reviewer DFUg**
>
> Thank you very much for your feedback on our work! We'd like to address some of your observations below:
>
> ### Missing comparison with cubic regularised Newton, trust-region Newton and their first-order sub-solvers
> Thank you for drawing our attention to the cubic-regularised and trust-region Newton methods – we agree these are relevant and will include them in our literature review. We will change the claim you highlight to more clearly isolate our specific contribution.
>
> ### No proof of convergence rate
> In Appendix D.3 we prove that our algorithm converges in regions of positive curvature and escapes regions of negative curvature, and Appendix D.4 analyses its rate of convergence, in both cases assuming infinitely many terms in our power series such that the exact saddle-free Hessian is implicitly used. Further, we argue in Appendix D.2 that a finite truncation of the power series prioritises the most impactful terms, and the initial $k=0$ term of the series in isolation gives precisely SGD with learning rate $\frac{1}{\sqrt{V}}$, which has been thoroughly studied for suitable choices of $V$. Since we apply our algorithm to ML problems which fundamentally differ from the convexity, smoothness and other assumptions generally used to motivate and analyse optimisation techniques, we feel this is a sufficient demonstration of our algorithm’s behaviour.
>
> ### Excessive hyperparameters and heuristics
> Firstly, we note that the series length $K$ and order of acceleration $N$ are largely ‘patience’ specifiers rather than hyperparameters, as it is clear that we should set these to the largest values we can tolerate. Secondly, we search over the same number of randomly-generated hyperparameter choices for each algorithm and plot results over time, so any practical issues with our algorithm would present in these results.  Lastly, we note that our method certainly uses fewer heuristics than K-FAC, which is nonetheless accepted as a second-order benchmark.
>
> ### Figure 3 dominated by K-FAC (Kazuki)
> In general, we feel it is better to report poor results fully than to omit them, especially in cases like this where the remainder of the plot, though compacted, is still legible. However, we could remove K-FAC (Kazuki) from the plot and simply note in the caption that it diverged if this would be preferable.

---

### Review · Reviewer_Ji7s · 2024-01-12

**Summary Of Contributions:**

This paper proposes computing a saddle free Newton type update via a matrix series expansion of the absolute Hessian instead of using direct matrix factorizations (e.g., Lanczos or eigenvalue decomposition). Both the proposed methods of this paper, as well as previous cited works can approximate the absolute Hessian matrix free where Hessian-vector products can be easily computed in automatic differentiation frameworks as $H(w)v = \frac{d}{dw}(g(w),v)$

The method is compared against other optimization methods such as Adam, SGD, KFAC, and other saddle free Newton methods, and shows comparable performance.

The appendices of the paper analyze the algorithm, giving a mathematical justification to the idea. Additionally convergence of the method and saddle point escape is analyzed.

**Audience:**

Yes

**Broader Impact Concerns:**

No concerns.

**Claims And Evidence:**

Yes

**Requested Changes:**

I would like to see a few things:

1. If possible some analysis of the rate of approximation of the Hessian by series expansion would greatly aid the central contribution of this work. And if this is infeasible, showing some numerical plots of this would be very interesting. This would really help illustrate the power of the method and make a much stronger case for publication.

2. I suspect that the optimal hyperparameters (e.g., large batch sizes) are masking some potential issues with the method. These issues are endemic to many Newton methods and not unique to this method. If the series expanded Hessian truly does solve the instability issues of earlier SFN type methods then the authors must demonstrate this numerically in the regimes where those methods had trouble: namely smaller batch sizes. These are also the typical setups for achieving good performance on datasets like CIFAR10 with architectures like ResNet. Please report generalization accuracy as a percentage, as this is the standard way of assessing the quality of a trained classifier. If the generalization accuracies are where they should be (e.g. >90%), this will give the reader more confidence in the method being practically useful. See for example this link: https://github.com/kuangliu/pytorch-cifar.

**Strengths And Weaknesses:**

Strengths: The paper is very well written. While the contribution is a bit incremental, it is elegant, and well motivated. The paper does a good job of summarizing other people's work. The authors present many different algorithmic considerations such as various hyperparameter optimizations and other accelerations that leads me to believe the numerical work is thorough.


Weaknesses: The idea is somewhat incremental, and essentially about approximating Hessians. I am not sure that a sequence of matrix-vector products is a practical means of approximating Hessians in modern machine learning computing. Can the mat-vecs all be done simultaneously? This would port better to GPUs and make a better argument for using an idea like this (which I would be more in favor of if these HPC considerations could be commented on).

The authors claim to essentially solve instability issues of earlier SFN works, but I do not totally follow the argument. Hessians of large neural networks are typically rank degenerate and require damping or some other means for their (approximate) inversion. The authors claim that the inverse of the absolute Hessian can be computed from a series expansion and this avoids the pitfalls of these earlier methods that are due to damping and low rank truncation. From my understanding of Algorithm 1 the authors are not using damping, which to me would seem to lead to instabilities, since the Hessian matrix is itself low-rank. However the appendices to list damping parameters for the method, so perhaps this can be articulated better in the main body to clarify this.

The numerical results should report generalization accuracies for the classification problems so that readers can understand if the regime the results are in is actually competitive for these large scale ML problems. For example, its well known that second order methods have better stability properties with larger batch sizes, this however often defeats the purpose as it leads to worse generalization accuracies. Small batch sizes tend to lead to better generalization accuracies, but also introduce additional statistical sampling errors that can cause instability in Newton methods.

---

> ### Author Response · Authors · 2024-01-23
> **Response to Reviewer Ji7s**
>
> Thank you very much for your feedback on our work! We'd like to address some of your observations below:
>
> ### Incremental contribution of impractically many matrix-vector products which can't be parallelised
> In its current form, the Hessian-vector products cannot be embarrassingly parallelised because we must compute term $k-1$ of the series before computing term $k$, and we rely on the gradient and Hessian evaluated at the current weights to perform this computation. However, we note that this additional cost penalises our algorithm when we quote results over runtime, and our results suggest that this isn’t a major issue.
>
> ### Claim to solve instability of SFN
> Our intention was not to claim a solution to the instabilities of SFN, but rather to present an alternative approach to SFN which avoids some of the hazards of the original algorithm (possibly while introducing additional hazards itself). This is what we tried to articulate in paragraph 6 on page 2, which we acknowledge reads like an overclaim, so we will change it to clarify this point. If your concern is sourced from elsewhere in the paper, we’d be grateful if you could please direct us to the text in question, because having revisited the paper, we believe this is the only place where we make any claims about stability which aren’t supported by experimental results.
>
> ### Unclear application of damping
> Thank you for pointing out this omission. For our unadaptive algorithm (denoted ‘Ours’ in the paper), we do in fact incorporate a fixed value of damping in the usual way, where we add a small multiple of the identity matrix to the Hessian at each use. Our hyperparameter search then selects the fixed damping value as described in Table 3. We will update our descriptions to make this clearer, and also to add missing scientific notation exponents in Table 3.
>
> ### Analysis of rate of Hessian approximation by series expansion (or numerical plots)
> While not phrased as a study of the evolving Hessian approximation, our comparison of series accelerators in Appendix B.5 illustrates the evolving update vector (approximate (SFN inverse-Hessian)-vector product) in terms of cosine similarity to and 2-norm of the exact vector. Further, our analysis of Appendix D.2 deconstructs our series calculation, showing the effective matrix by which we multiply the gradient $\mathbf{g}$ has precisely the eigenvectors of the Hessian (and its inverse), with eigenvalues evolving according to independent series computing $+\sqrt{\lambda^{2}}$ for each eigenvalue $\lambda$. Together, these studies give a sense of our series’ convergence behaviour.
>
> ### No reporting of test accuracies for classification tasks
> While we acknowledge the interest in accuracy metrics, we would expect the evolution of these to be very similar to the test losses we already report. Unfortunately, test accuracy is not one of the metrics saved in our experimental logs and so we would need to rerun all of our experiments to be able to provide accuracy metrics. We believe the incremental benefits of reporting accuracies to the reader wouldn’t justify the cost of re-running all our experiments. However, the inclusion of tuned trajectories for SGD and Adam should adequately illustrate the performance we would typically ‘expect’ from these model and dataset combinations.
>
> ### Not reporting test error may mask a poor trade-off between generalisation and stability, a known second-order problem
> We acknowledge the concern that larger batch sizes mask instabilities in second-order methods and harm generalisation, but there are several mitigations for this. Firstly, our hyperparameter selection strategy optimises for validation performance, so should be capable of rejecting extreme batch sizes which cause instability or very poor generalisation. Secondly, having chosen a batch size, we plot the evolving test loss alongside the corresponding training loss, which then clearly illustrates any tendency to generalise poorly, and we see our algorithm performs comparably to SGD and Adam, suggesting any generalisation penalty is not substantial. Ultimately, we handle these risks by deliberately surfacing them in hyperparameter search and evaluation, and we contend that a configuration which demonstrates adequate stability and generalisability is of interest regardless of its batch size.

---

### Author Response · Authors · 2024-01-23
**Updated Paper Posted**

We have updated our submission in response to reviewer feedback, with the following changes made:

* Claims about our algorithm have been adjusted to more precisely describe its novelty (Abstract; paragraph 3 of Section 1; and paragraph 1 of Section 5)
* Additional literature review added to Section 2 describing other non-convex optimisation methods (paragraph 5); the stability of the Lanczos algorithm (paragraph 6); other saddle-avoiding methods (paragraph 7); and the connection to LiSSA (paragraph 8)
* Updated description of our algorithm in Section 4.1 to include our use of fixed damping
* Corrected missing scientific notation in Table 3

---

### Decision · Action_Editor_dCNj · 2024-02-21

**Recommendation:** Accept with minor revision

**Comment:**

The paper introduces an approximate Newton method that makes uses a Taylor series expansion of the matrix square root. This avoid issues with negative eigenvalues, and it does not require computing an SVD.

Furthermore the paper is well written, the claims are clear and substantiated, and now with the expansion of the background to include cubic Newton, trust region methods, saddle-avoiding methods, Lissa ..etc I also find the background to now be good. I agree with DFUg that reporting test accuracy (instead of loss) would be helpful. But since you make no claims of having a SOTA test accuracy on these benchmakrs,  I think this omission is reasonable.

Overall, because of this novelty, clarity and good background, I recommend the paper be accepted.

I also find reading Figure 2 difficult because of the divergence of one of the KFAC method. May I recommend that you make use of extra space in the camera ready to duplicate this figure, one with and one without this divergence KFAC method

**Audience:**

This paper is of interest to the optimization community, in particular those trying to develop a scalable stochastic second method that can be applied in the non-convex setting.

**Claims And Evidence:**

The paper claims to make use of a new series expansion of the matrix absolute value with a Newton method. Neither, or the reviews have seen this be used before, and the paper has now an extensive background. The paper also claims their method can scale to reasonable dimensions, and is competitive with first order methods both in terms of runtime and steps. Their numerics on Fashion-MNIST, SVHN and CIFAR-10 data sets justify this claim.